# Internality and the internalisation of failure: Evidence from a novel task

**Federico Mancinelli**[1,2,3]*, **Jonathan Roiser**[4], **Peter Dayan**[1,5,6]*

**1** Gatsby Computational Neuroscience Unit, University College London, London, United Kingdom, **2** Centre for Computation, Mathematics and Physics in the Life Sciences and Experimental Biology (CoMPLEX), University College London, London, United Kingdom, **3** Department of Cognitive Neuroscience, Scuola Internazionale Superiore di Studi Avanzati, Trieste, Italy, **4** Institute of Cognitive Neuroscience, University College London, London, United Kingdom, **5** Department of Computational Neuroscience, Max Planck Institute for Biological Cybernetics, Tübingen, Germany, **6** University of Tübingen, Tübingen, Germany

* federico.mancinelli@sissa.it (FM); p.dayan@ucl.ac.uk (PD)

## Abstract

A critical facet of adjusting one's behaviour after succeeding or failing at a task is assigning responsibility for the ultimate outcome. Humans have trait- and state-like tendencies to implicate aspects of their own behaviour (called 'internal' ascriptions) or facets of the particular task or Lady Luck ('chance'). However, how these tendencies interact with actual performance is unclear. We designed a novel task in which subjects had to learn the likelihood of achieving their goals, and the extent to which this depended on their efforts. High internality (Levenson I-score) was associated with decision making patterns that are less vulnerable to failure. Our computational analyses suggested that this depended heavily on the adjustment in the perceived achievability of riskier goals following failure. We found beliefs about chance not to be explanatory of choice behaviour in our task. Beliefs about powerful others were strong predictors of behaviour, but only when subjects lacked substantial influence over the outcome. Our results provide an evidentiary basis for heuristics and learning differences that underlie the formation and maintenance of control expectations by the self.

## Author summary

How success in tasks depends on our efforts is largely a feature of what is known as the controllability of the situation or environment. This quantity should determine the way we approach, adapt to, and perform tasks. In novel settings, it can only be our expectations about controllability that exert an effect, for instance determining the balance of our focus between features of achievability and potential reward; or affecting prior notions about what exactly is achievable or what we can or cannot do. All of these might be different between aversive and appetitive domains. To study these issues, we designed a novel task in which subjects have to learn about a new environment, and analyzed their behavior using a rich computational model. We found that expectations about controllability played a particularly important role in influencing learning, but in a way that differed between positive and negative outcomes. In particular, high expectations about controllability were tied to higher learning rates of unachievability given negative outcomes,

**Data Availability Statement:** Data availability: https://github.com/fedmanci/control-task-reproducibility-materials.git.

**Funding:** F.M. received financial support from The Engineering and Physical Sciences Research Council (through CoMPLEX at University College

London) and The Gatsby Charitable Foundation; J. R. received support from the Wellcome Trust; P.D. received support from the Gatsby Charitable Foundation, The Max Planck Society, and The Alexander von Humboldt Foundation. The funders had no role in study design, data collection and analysis, decision to publish, or preparation of the manuscript.

**Competing interests:** The authors have declared that no competing interests exist.

guarding subjects from further loss, and preserving their subjective optimistic expectations about control. Our findings can be interpreted within a theoretical framework which implicates control expectations in individual learning differences, and fit well within modern theories of learning in aversive contexts and serotonergic function.

## Introduction

The controllability of a particular environment quantifies the extent to which the actions that we can choose in that environment take us adequately quickly, reliably and cheaply to desired outcomes such as the acquisition of rewards and the avoidance of punishments. Uncontrollable environments may be temporarily benign—but for our reactions to new opportunities and threats to be successful, controllability is essential.

Versions of this objective notion of controllability underpin work in areas such as learned helplessness (when animals learn that one environment is uncontrollable, and generalize this to new environments) [1, 2]. Learned helplessness is used as a model of forms of depression, leading to normative insights such as that uncontrollable environments aren't worth exploring since they aren't reliably exploitable. Attempts have therefore been made in the field of neural reinforcement learning to look in a more granular manner at formalizing the separate components of controllability in terms of prior expectations about the environment [3].

However, it is the subjective perception of controllability that has the ultimate psychiatric import, and can diverge from the objective facts according to the ascription of responsibility for successes and failures (see, e.g. [4]). This subjective perception is captured by scales such as the locus of control (LoC; [5]). People with a so-called *external* locus of control might, for instance, ascribe failure to the vicissitudes of bad luck or evil underlying forces and are distinguished from those with an *internal* locus of control, who believe that they are responsible for outcomes, and so think that their internal incompetence must be at fault. Such ascriptions have an important, and complex, knock-on effect on learning, since, for instance, if chance can be blamed for failures, and skill for successes, subjects will build a biased impression for themselves of the controllability of the environment [6–8]. Further, people with an external locus of control might also rationally conclude that behavioural adaptation will not be necessary to obtain better performances. Though LoC is usually thought of as a form of personality orientation, external loci are a common co-occurring trait in various disorders—and shifts towards internalization are often a consequence of treatment (see [9] p.42). In a meta-analysis, [10] found solid relationships linking external orientations with depression severity, as measured by a variety of questionnaires (e.g. BDI); and generally, the number of studies which evidence an inverse relationship between locus of control and psychiatric symptoms is quite large (see, e.g. [11–13]). Most human studies attempting to individuate the behavioural implications of LoC and attribution have done so by having different subject groups fill in questionnaires and measuring difference in their responses (see, e.g. [14, 15]), or framing tasks as being determined by skill/chance [5, 16–18], or using gambling or risk-aversion paradigms [19]. However, simple gambling-based paradigms fail to offer a comprehensive picture of controllability in at least two ways. First, they provide no purchase on the scale of controllability. For instance, imagine having to ride your bike to reach as swiftly as possible to a shop before it closes. Would you rather use a bike you know to be really guidable and swift, but parked further away from the shop—or use a rather old, and therefore somewhat defective, bike which is parked closer? if these two are placed so that the overall probability of reaching the shop in time is the same (macroscopic control), then someone who prefers microscopic control will be more

inclined to use the farther bike (since the effort invested in driving it translates more noise-lessly to covering distance). Second, gambling is explicitly styled in terms of luck, reducing the scope for skill or competence to play an important role.

Here, we present a more ecological task intended to separate different components of controllability, allowing us to discriminate the subjective sensitivity to some of its micro- and macro-scopic aspects, and their interaction with outcomes, ascription style, expectations and choices. In our task, subjects choose between, and then use, one of a range of tools to achieve various simple goals. The tools are vehicles (as in our example of the bike), and are only partially reliable (with differing amounts of entropy in the effects of choices; a quantity we will refer to as guidability) and intermittently effective (affording many or few actions per unit interval; we call this influence), to degrees that need to be learned. It is the requirement for execution that makes the subjects' achievements depend at least partly directly on their microscopic behaviour, and, as above, that makes the learning of the probability of achievement dependent on ascription style (did I fail because I did not press hard enough? did I fail because of bad luck?).

More specifically, our manipulations are aimed at resolving how internality (I-scores in the LoC scale; [14]) and beliefs in the influences of Luck (C-scores) (1) affect the choice between plans offering various trade-offs of macroscopic controllability and reward, (2) are manifest in explicit biases towards either macroscopic or microscopic scales of controllability, and finally (3) affect the way outcomes (in the form of success and failure to achieve a goal) guide the learning of controllability and, in turn, decision making. Previous work touching on these issues is limited (see, e.g. [20, 21])—and we know of no previous paradigm in which subjects had to learn controllability as they performed actions to reach goals.

The study that perhaps comes closest is by Julian et al., who reported a dart throwing paradigm in which subjects first estimated the two distances from which they could score with five and seven darts respectively (this step equates the macroscopic controllability of the two options), and were subsequently asked to choose the one distance from which they would prefer to throw (i.e. five darts from the closer, and seven darts from the farther distance) [20]. The closer distance offers higher microscopic controllability, since each single throw has a higher probability of scoring—and this is the option that internals were found to prefer. However, by design of the paradigm, (1) subjects do not learn from their own performance, and can not revisit their choices based on their successes and failure, and (2) the trade-off with macroscopic controllability is not manipulated (what if, for instance, subjects were given more darts to throw from the farther distance—how many more darts would it take to switch the preference to the farther set of tries?)—so that we can not examine the influences of attribution on learning, and the tradeoff between macro- and micro-scopic controllabilities. We sought to design a more comprehensive paradigm addressing these issues, and allowing us to examine how interactions among prior expectations, attributions, and learning, affect choice and then performance.

Our primary finding was that, in both conditions of high and low influence, internality with respect to self (Levenson I-score; [14]) was inversely correlated with choices of less attainable, more rewarding goals. In good influence conditions, where efforts are predictive of outcomes, internal subjects prioritized features that are tied to achievability, and in doing so managed to lose on fewer trials. In both good and poor influence conditions, internality was strongly tied to the speed of learning from failure that the more rewarding goals would not be achievable. High I-scorers learned unachievability faster, whilst low I-scorers perseverated in attempting the next-to-impossible, more rewarding goals. Lastly, we found serendipitously that beliefs about powerful others were strongly predictive of performance in low influence conditions.

## Methods

### Ethics statement

The study was approved by the UCL Graduate School Ethics Committee (ethics ID: fmri/ 2013/005).

### Participants

We recruited 37 subjects through the Institute of Cognitive Neuroscience subject database at University College London (21 females; age, median: 24, highest: 39, lowest: 18 yrs). All subjects completed a structured telephone interview to confirm the absence of any previous or current mental disorder. On arriving in the lab, subjects approved and signed an ethics consent form, and were instructed to read a briefing sheet which explained the task. Before starting the task fully, subjects performed 3 practice trials in both high and low influence conditions (see Experimental design), under supervision of the experimenter; they were told that they could ask for clarification if they were unsure about any aspect of the experiment, and that they could begin when they were ready to do so. After performing the task, subjects completed Levenson's LoC questionnaire [14]. They then collected a payoff which consisted of the sum of £10 and the actual outcomes of 30 trials sampled at random. Subjects knew the structure of the payoffs. Two female subjects were excluded when performing data analyses. One subject was excluded on account of having too many irrational vehicle and goal choices on *Catch* trials (more than 4 standard deviations away from the mean of the remaining subjects). This may be indicative of poor understanding or lack of focus throughout the task. A second subject was excluded due to an outlying low I-score (more than 3 standard deviations away from the mean of the remaining subjects). These exclusions left us with 35 subjects.

### Questionnaire

The perception of behavioural control is traditionally measured using questionnaires. Rotter introduced the "I-E" (Internal-External) forced-choice scale [5]. Here, we adopted a more finely graded, Likert type scale [14]. This dissects LoC into three separate dimensions: "Internality" (I), "Chance" (C), and "Powerful Others" (P). In her work, Levenson motivates this choice: "The rationale behind this tripartite differentiation stemmed from the reasoning that people who believe the world is unordered (chance) would behave and think differently from people who believe the world is ordered but that powerful others are in control. [···] Furthermore, it was expected that a person who believes that chance is in control (C orientation) is cognitively and behaviourally different from one who feels that he himself is not in control (low I scale scorer)".

In a further difference from Rotter's original scale, the statements underpinning Levenson's assessments are phrased so that it is clear that they pertain only to the person answering. That is, they measure the degree to which an individual feels she has control over what happens, rather than what she feels is the case for "people in general". Levenson also ensured that the LoC measures were not influenced by demand characteristics, showing that there was no significant correlation between her scores and the Marlowe-Crowne Social Desirability Scale [22]. In Table 1 we give a general overview of our analyses concerning the Levenson LoC questionnaire in our sample. The analyses consisted of (1) a VariMax rotated factor analysis, to substantiate the claim that responses can be accounted for by three dimensions [23] and (2) computation of internal consistency measures for each subscale.

**Table 1. Analysis of LoC subscales.** In this table we report various measures to gain insight into the structure and consistency of the LoC subscales. We performed a varimax-rotated factor analysis on the single items to obtain the factor structure of the questionnaire responses. We used four components ($p_{\chi^2} = 0.1$). The cumulative variance explained by the first three factors was 35%, in line with the values found in literature (see, e.g. [23]). We correlated scores (i.e. the projections of the questionnaire responses along the factors) with the I,P and C scales to see whether these were reflected (coefficients significant post false discovery correction for multiple comparisons are in bold). The results replicate well those found in literature, as they suggest that the first three factors largely reflect the 3 dimensions individuated by Levenson (Levenson 1974). Although the fourth factor appears to individuate a further, non-specific external component (as it is anticorrelated with I-scores, and weakly positively correlated with C and P scores) none of its correlation coefficients reach significance, which suggests that the variability that this dimension captures is not enough to warrant adding a further dimension. Finally, we report internal consistency measures, along with 95% confidence intervals, in the last column.

| Levenson subscale | Factor loadings | | | | Internal consistency (Cronbach's $\alpha$) |
|---|---|---|---|---|---|
| | factor 1 | factor 2 | factor 3 | factor 4 | |
| Internality (I) | −0.47 | −0.29 | **0.78** | −0.46 | **0.73**, 95%CIs [0.57, 0.88] |
| Chance (C) | **0.64** | 0.40 | −0.36 | 0.28 | **0.79**, 95%CIs [0.67, 0.91] |
| Powerful Others (P) | 0.41 | **0.77** | −0.47 | 0.26 | **0.79**, 95%CIs [0.67, 0.90] |

## Power analysis

Our analyses aim at resolving the individual relevance of LoC subscales in our task, in which we hypothesise Levenson I- and C- scales to play substantial roles. With 35 subjects, we had a power of 0.8 to detect correlation coefficients of $\sim 0.45$, with a two-tailed significance threshold of $p = 0.05$.

## Experimental design

The task consists of a novel videogame (coded in JavaScript). On each trial, subjects see a large blank canvas, and have to select between, and then move, one of two vehicles (shown as small circles) to a goal (large circle) within 14 seconds by pressing the arrow keys on a keyboard. Movement is only allowed horizontally and vertically, and in predefined step-sizes. If subjects are successful, they win a reward whose magnitude (in pence) is signalled at the goal. If they fail, they lose a fixed amount of £0.15. Vehicles differ in their placement on the canvas, and in the extent of control they afford (guidability). A vehicle's guidability simply defines how likely it is to move in the direction implied by the arrow key pressed by the subject as opposed to moving in any cardinal direction (chosen apparently uniformly at random). Subjects learn about the characteristics of each vehicle from the experience gathered within a block. On some trials, subjects are offered two potential goals (double goal trials; DGTs); on others, there is only one (single goal trials; SGTs). Goals differ by the amount of reward, and, in most trials with more than one goal, the one that is more rewarding is further away than the one that is less rewarding. Trials differ in registering more ('high influence') or fewer ('low influence') actual moves per second, though subjects can always press keys as fast as they like. In order to avoid potentially confounding fatigue effects, participants are offered the opportunity to rest at the end of every trial (by adding 10 seconds to the inter-trial interval) before the next trial starts. Fig 1 illustrates the timeline of a trial with two goals.

**Rationale of the design and control manipulations.** The gross probability of attaining a goal in our task depends on (a) the *distance* separating the vehicle from the goal—a significant contributor to the macroscopic controllability of that goal (i.e. the overall probability of reaching the goal); (b) the *guidability* of each vehicle, by which we operationalise microscopic controllability; and (c) the *influence* that subjects can exert, which sets the efficacy of effort.

We parameterised the guidability of the vehicles in terms of conditional probability [3]: independently, at each press, the vehicle follows the subject's intended direction (i.e., the chosen arrow key) with probability $\gamma$, or moves in a random direction (drawn uniformly) with probability $1 - \gamma$ (note that this includes the arrow direction followed by chance). For a certain pressing frequency, the true probability of success depends both on how (1) distant from the

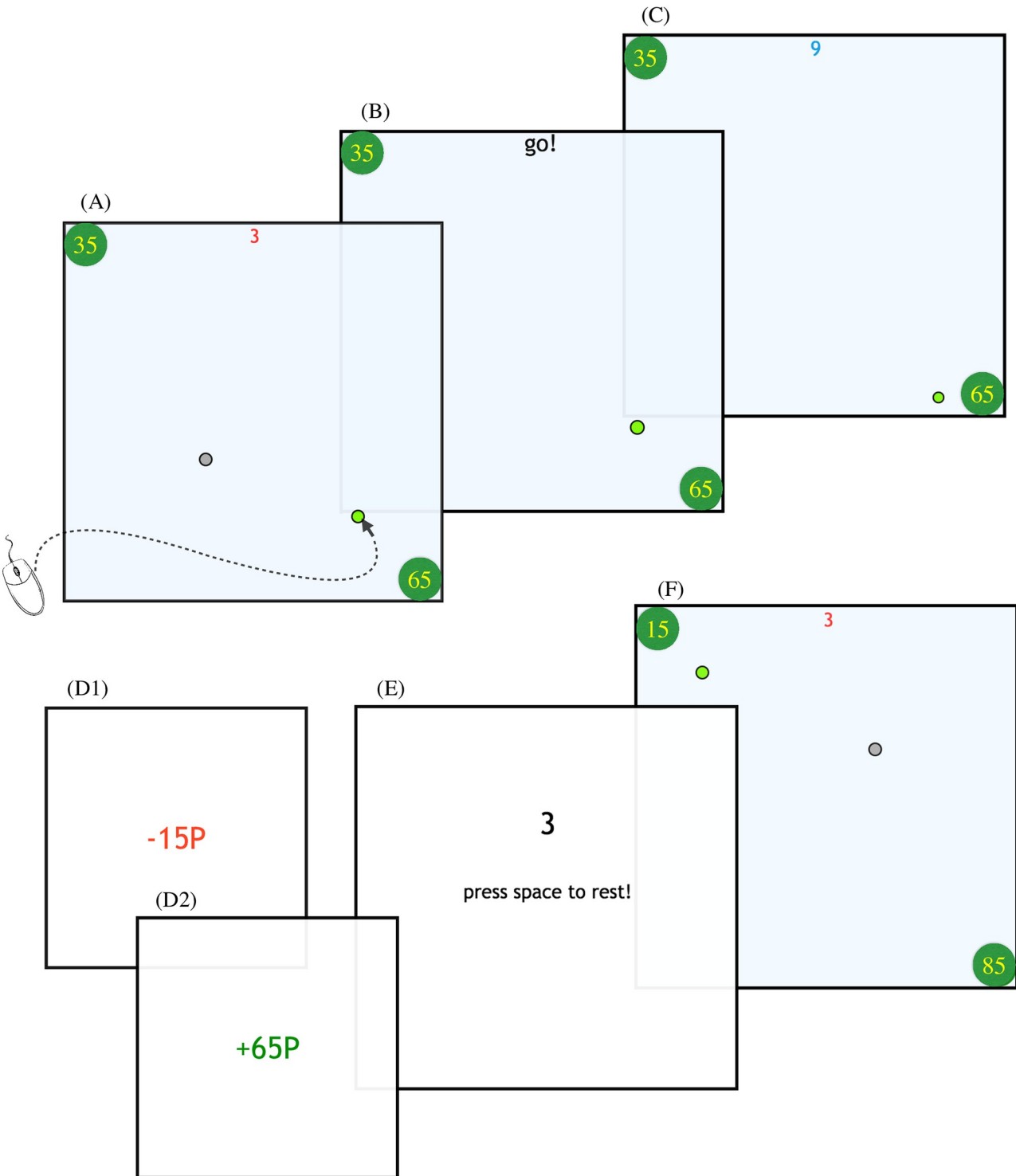

**Fig 1. The timeline of a trial.** This trial is of type *OG* (see Fig 2). This is the only trial type where $r(g^-)$ is larger than $r(g^+)$, intended to prevent subjects from starting to assume that the larger goal would always be further away in DGTs. From top-left to bottom-right: (**A**) vehicle selection phase. Subjects had 5 seconds (3 seconds of which are left in the figure) to choose a vehicle using the mouse pointer (whose movement is shown by the curved arrow). The ultimate goals are shown as green circles whose values in pence are shown as numbers inside each. (**B**) Any unchosen vehicle is removed, and a "go!" cue starts the movement phase of the trial. The subject then had 14 seconds to drive the chosen vehicle to reach one of the goals to win the amount shown. Otherwise, they lost 15p. (**C**) the trial is in progress—in this case with 9s left (top centre) for moving. (**D1**) shows what subjects see upon expiry of the allowed time; the amount lost (15p) is displayed in red, and is accompanied by a noxious sound. (**D2**) shows what subjects see if they touched the

goal before the time elapsed. The amount won is displayed in green accompanied by a pleasurable sound. (**E**) waiting for the next trial to start (3s). At the end of the twentieth trial of each block, participants saw a sign saying "vehicles change", which notified the end of the block. They then had a 5s rest before starting again. On pressing the space bar, 10s would be added to the count-down. (**F**) The new trial (of type $t_1$) begins.

goal, and (2) guidable the vehicle is, and is nearly always (except for trial type *OG* in double goal trials) orthogonal to the amount of reward that the goal offers—so that the sensitivity to macroscopic controllability can simply be measured as the proportion of choices to attain the less rewarding goal. Analogously, microscopic control sensitivity can be measured as the proportion of choices of the more controllable of the two vehicles. Subjects preferring lower rewards are more sensitive to macroscopic control than reward (since, in DGTs, distances and reward amounts are orthogonal); conversely, subjects preferring more rewarding goals regardless of the vehicle used neglect either form of control; and finally, subjects preferring more controllable vehicles are more sensitive to microscopic control. The manipulation of influence is perhaps the most restrictive—as it sets a hard limit on the maximum number of steps per second that vehicles would move. We signaled two types of block: a benevolent type ("high influence") in which this limit was set to 8 steps per second (8*Hz*), and a malevolent type ("low influence") in which this limit was set to 4*Hz*. Note that all subjects could reach (and in many cases, break) the maximum pressing frequency of 8*Hz*. In high influence blocks, higher pressing frequencies are designed to entail higher rewards, but in low influence blocks, the two are uncorrelated. We introduced this manipulation to test whether influences of internality on behaviour would be more salient in inherently benign or malignant environments (i.e. environments in which higher efforts lead to higher rewards on average, and in which they do not, respectively).

In order to maximise expected rewards, subjects should approximate the probability of reaching the goal with a certain vehicle, combining distance and guidability information (i.e. how likely a vehicle with a certain guidability is to cover the distance separating it from the goal in a timely manner). In forming this approximation of probability, subjects will not just use trial features such as distance and guidability—their choices will be informed by the history of achievement and failure (which is subject to influence), and thus on their subjective attribution schemes. For instance, if I failed to reach the goal with a certain vehicle from a certain distance, the question of whether I will try again should at least in part be informed by my prior experience of failure. As we will see, our task also allows disentangling such history-dependent influences through simple model-based analyses.

**Task structure.** The full task comprised 160 trials, divided in 8 blocks. Within each block, the first 16 trials included two goals (DGTs). Four single goal trials (SGTs) appeared at the end of each block, by which time we assumed that subjects would have a good grasp of the controllability of each of the two vehicles. The design for both DGTs and SGTs is illustrated in Figs 2 and 3. The influence condition, or block type ("high" or "low" influence) was manifest in the background colour of the canvas, either light beige or blue (counterbalanced across subjects). Subjects were instructed as to the link between background color and influence.

DGTs (128 trials per subject) always included two vehicles and two goals. At the start of each block, subjects have no knowledge about the worth of each vehicle, and must learn in order to know what they can, and cannot, do. All blocks involved all the DGTs (8 trial types) indicated in the Table to the right of Fig 2 in random order. The same trial appears twice, (1) with a randomly rotated canvas (by multiples of 90 degrees), and (2) with vehicles swapping positions; this is in order to obtain a fully orthogonal design. In DGTs, there is always a vehicle at the center of the canvas, so that subjects face the recurrent question, at each trial, of whether they will be able to achieve the more rewarding goal using the vehicle currently at the center;

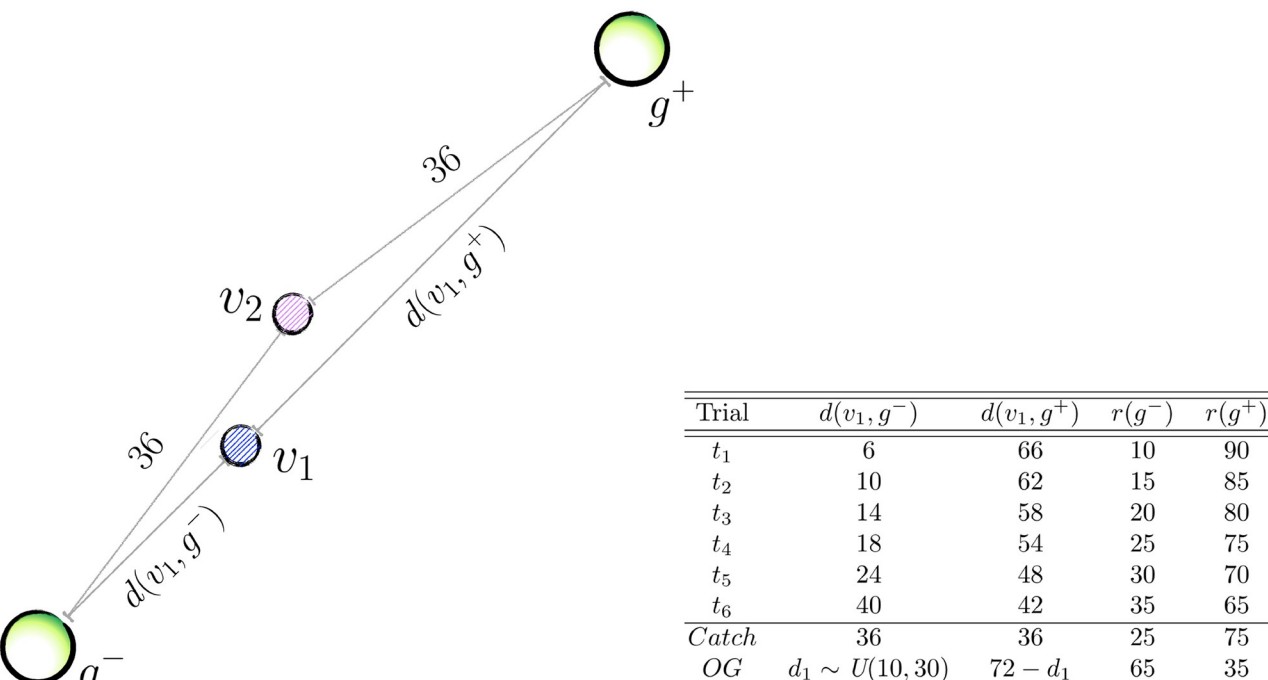

| Trial | $d(v_1, g^-)$ | $d(v_1, g^+)$ | $r(g^-)$ | $r(g^+)$ |
|---|---|---|---|---|
| $t_1$ | 6 | 66 | 10 | 90 |
| $t_2$ | 10 | 62 | 15 | 85 |
| $t_3$ | 14 | 58 | 20 | 80 |
| $t_4$ | 18 | 54 | 25 | 75 |
| $t_5$ | 24 | 48 | 30 | 70 |
| $t_6$ | 40 | 42 | 35 | 65 |
| *Catch* | 36 | 36 | 25 | 75 |
| *OG* | $d_1 \sim U(10, 30)$ | $72 - d_1$ | 65 | 35 |

**Fig 2. Illustration of double-goal trials (DGTs).** Vehicles' colours in this figure are random. $d(v_1, g^-)$ and $d(v_1, g^+)$ indicate the Manhattan distances (in units of the vehicle's step size) from vehicle $v_1$ to both $g^-$ and $g^+$. Vehicle $v_2$'s distance from both goals is constant, i.e. $d(v_2, g^-) = d(v_2, g^+) = 36$. The last two columns indicate the monetary reward (in pence) associated with each goal, i.e. $r(g^-)$ and $r(g^+)$. The theme underpinning the design is, the closer a vehicle is to $g^-$, the less rewarding $r(g^-)$ will be, and the more rewarding $r(g^+)$ will be. In this illustration, we arbitrarily placed $v_1$ asymmetrically (closer to $g^-$), and $v_2$ at the center of the canvas. Recall, however, that all trials showed in the Table appear twice in every block, the second time with vehicles' positions swapped. This was done to ensure that we can fully orthogonalise rewards, distances and guidabilities. Thus, if we first encounter trial $t_4$ with $v_2$ at the center of the canvas (as is shown), the second time we meet this trial type the vehicles' positions will be swapped (and the canvas randomly rotated) and $v_2$ will be the one closer to $g^-$. *Catch* and *OG* trials are atypical, and test participants' notion of the vehicles' guidabilities, and break the inverse correlation of distance and reward imposed by the design of the other trials, respectively. In *Catch* trials, vehicles are equidistant from both goals, so we expect participants to choose the more guidable vehicle ($v_1$) and the more rewarding goal. Obvious-goal (*OG*) trials are the only occasion in which $r(g^-)$ is actually larger than $r(g^+)$, so that choice should "obviously" be $g^-$. The distance of the closer vehicle to the 65p goal is extracted from a uniform distribution in the indicated set (we used a MatLab notation to denote the sample space); this is the only occasion in which distances are sampled randomly within the design. All DGTs are individually rendered, for clarity, in S1 Fig, where we show a full instance of the first 16 trials in a block. An *OG* trial is also shown in Fig 1.

as we will see, our computational analyses will shed some light on the learning process underlying the formation of this judgement. In S1 Fig, we illustrate an instance of how the 16 DGTs might be laid out in a block.

SGTs (32 trials in total per subject) were explicitly designed to pit goal distance against vehicle guidability, to reveal whether subjects might exhibit a bias towards either, or whether they would choose approximately optimally, according to the probability of reaching the goal. SGTs were, on average, harder in both influence conditions. This is because the furthest vehicle from the single goal is at a distance larger than that between the central vehicle and the more rewarding goal in DGTs. Throughout SGTs, the more guidable vehicle is always at a distance of 44 Manhattan steps from the goal; the less guidable one lies instead at a closer distance to the goal which is a direct function of the fraction of the vehicles' guidabilities. This heuristic was used to guarantee that in two out of 4 SGTs, the best vehicle to use would be the closest to the goal, and in the reamaining two, the farthest.

Within each influence condition, the four blocks differed in the guidability of the pair of available vehicles. In one block, the two vehicles would be equally guidable, and their guidability level would be "Medium" (see Fig 4 for the values of guidability of each level); in the

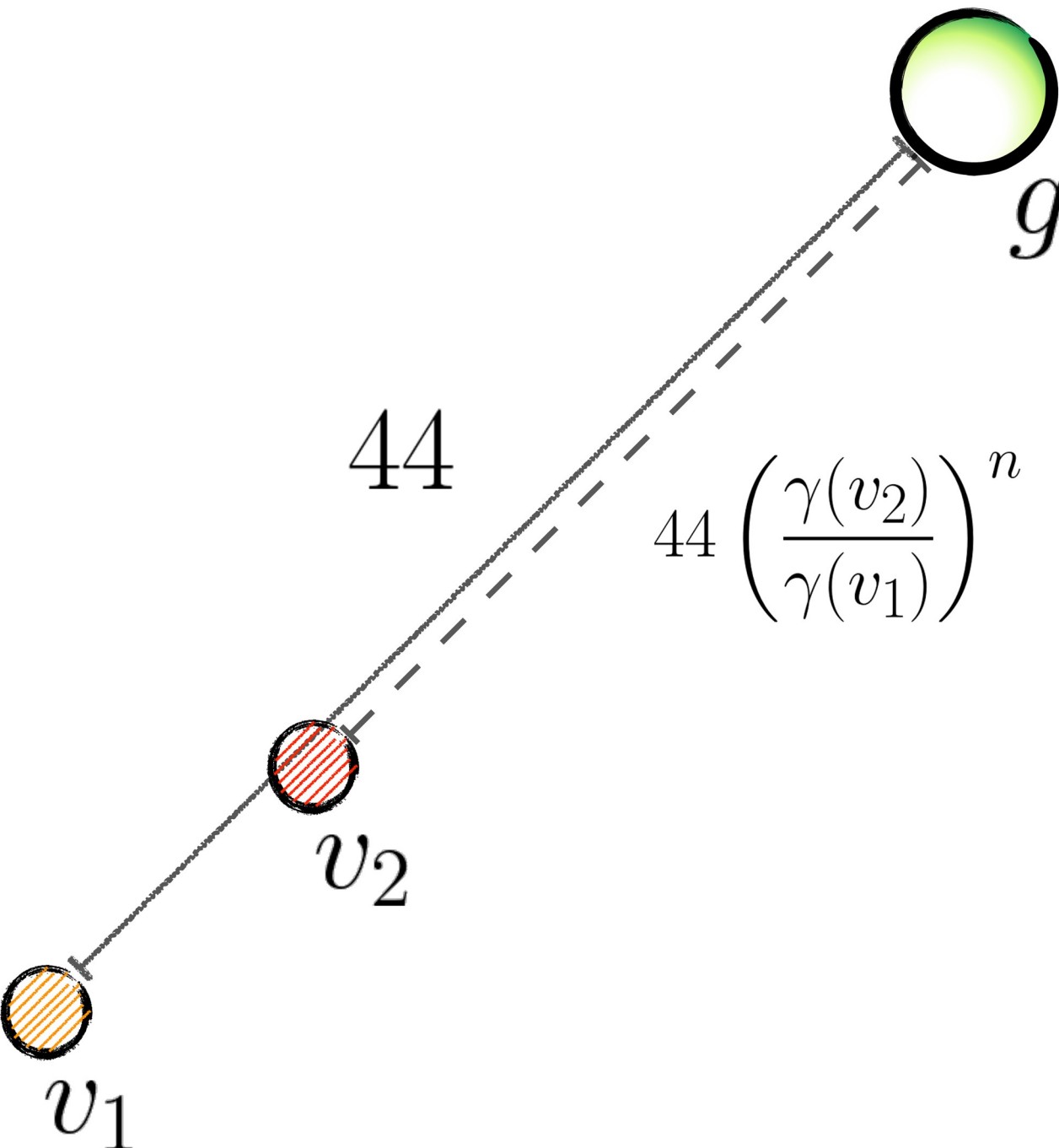

**Fig 3. Illustration of single-goal trials (SGTs).** Vehicle's colors in this figures are random. In SGTs, the vehicles' relative positions from the goal ($g$) depend on their guidability. The more controllable (yellow, in figure) vehicle's distance from $g$ is always 44 Manhattan steps. The other (red) vehicle's distance is instead a fraction of this distance equal to powers (the exponent $n$ takes the values 1,2,3 and 4) of the ratio of the guidabilities, i.e. $\gamma(v_2)/\gamma(v_1)$. Setting the distance of the red vehicle from $g$ to be dependent on this ratio promotes harder decision making; and we empirically verified that it ensures that $v_1$ is the correct choice exactly half of the time if subject pressed at a frequency of $8Hz$ (in high influence conditions, it is best to choose the most controllable vehicle when $n < 3$, and the closer vehicle when $n > 2$). Thus, choosing the same vehicle on all four SGTs is then wrong. If the two vehicles are equally controllable, their distance from the goal is randomly 44 and 38 Manhattan steps, and of course the closest vehicle to $g$ is always the right choice. The probability of making the goal in low influence conditions is instead hopeless, by design (no win was recorded). Rewards associated with the single goal are pseudorandomly distributed in a uniform interval between 50 and 80 pence, in steps of 10. The order of the four SGTs is randomized, but they always occur at the end of each block, when the guidability of each vehicle can be assumed to be known by participants.

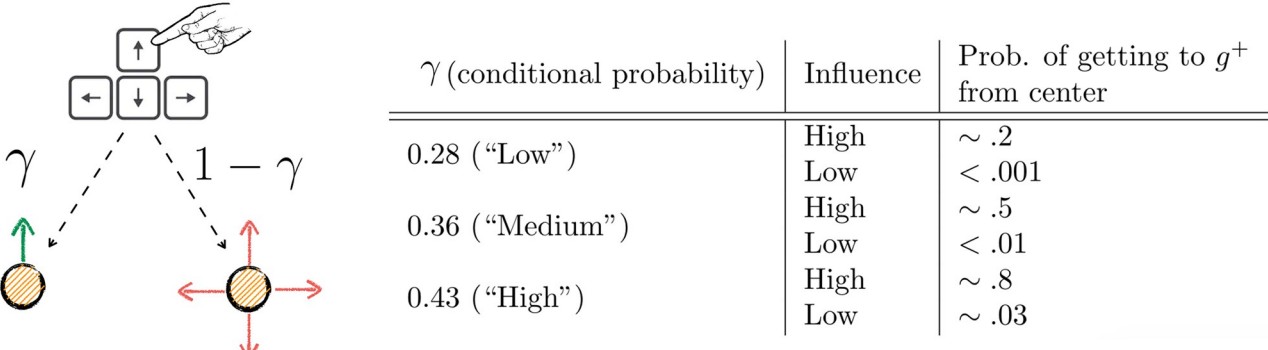

| $\gamma$ (conditional probability) | Influence | Prob. of getting to $g^+$ from center |
|---|---|---|
| 0.28 ("Low") | High | $\sim .2$ |
| | Low | $< .001$ |
| 0.36 ("Medium") | High | $\sim .5$ |
| | Low | $< .01$ |
| 0.43 ("High") | High | $\sim .8$ |
| | Low | $\sim .03$ |

**Fig 4. The design of guidability.** The figure on the left hand side depicts the relationship between miroscopic actions and outcomes with respect to the conditional probability $\gamma$. On pressing (for instance) the upwards arrow key, the vehicle (small yellow circle) follows the direction verbatim with probability $\gamma$ (left; green arrow) or goes in a direction sampled uniformly at random (right; red arrows). On the right hand side, a table describes what the three different values for $\gamma$ imply when a vehicle placed at the center of the canvas is set to reach $g^+$, according to whether influence conditions are high or low. Reaching $g^+$ from the center of the canvas in low influence conditions is almost impossible.The choice of guidabilities implied substantially different probabilities of reaching a goal from the central position (using the central vehicle) in high influence blocks. For instance, pressing consistently at a frequency of $8Hz$ for the full duration of the trial (14s) would yield probabilities 0.8 ("High"), 0.5 ("Medium"), and 0.2("Low"). In low influence blocks, the chance of making a goal using the central vehicle was circa 0.03 ("High" guidability) and fell substantially below 0.01 when guidability is "Medium" and "Low". These numbers are Monte Carlo estimates obtained via empirical simulations, which assumed an optimal strategy of pressing, consisting of moving the vehicle only in the direction of the goal in the allotted time, at the maximum frequency allowed (8 or 4 Hz according to influence condition). Note that while we used empirical probability of reaching the goal from the central position to inform the value of guidability, we did not use it to determine the relative positions of the other vehicle.

remaining three, they would be faced with a ("Medium", "High") pair, a ("Low","Medium") pair, and finally a ("Low","High") pair. Vehicles could always be identified through their colours within a block, but their associated guidabilities had to be learnt from the start of each new block. The design choices of the conditional probabilities to assign to vehicles were based on the fine tuning of the empirical odds of reaching $g^+$ from the central position of the canvas. This process produced values for $\gamma$ which were used during the main experiment as per Fig 4.

**Notation.** Throughout the text, we denote the two vehicles in a block as simply $v_1$ and $v_2$. Recall that vehicles are identified by their colours, and differ in their guidability values (except for one block per influence condition where both of them have "Medium" guidabilities). We denote vehicle $v_i$'s guidability (as inferred by trial $t$) as $\gamma_t(v_i)$, and its true guidability as $\gamma(v_i)$. As SGTs are faced after 16 trials in each block, we drop the pedix $t$, and safely assume subjects know the true guidability of the vehicles. Without loss of generality, we will always assume that $v_1$ is the more guidable vehicle. We will specify that $v_1$ and $v_2$ are equally guidable when this is the case (i.e. both have "Medium" levels of the guidability). Finally, again in DGTs, we will use an auxiliary variable ($c_t$) to indicate which vehicle is at the center of the canvas at trial $t$, so that, for instance, $c_t = 1$ indicates that $v_1$ is at the center of the canvas. Recall that each of $v_1$ and $v_2$ is placed at the center of the canvas an equal number of times through the DGTs in a block.

A goal will simply be denoted $g$, and $r_t(g)$ the reward connected to it at trial $t$. We will add an apex as a shortcut to signalling that one goal is the more rewarding one in a DGT, so that $g^-$ signals the less, and $g^+$ the more rewarding goal. For SGTs, where there is only one available goal, we will simply write $g$.

## Model agnostic analyses

We examined the relationship between I- and C- scores and our outcome measures. We had no hypotheses concerning P-scores, so exploratory analyses on these data are reported separately. All our model-agnostic analyses were performed in each influence condition separately.

In DGTs, maximising rewards requires learning the achievability of goals via using the vehicles available in the block at hand. Nevertheless, choosing the more rewarding goal ($g^+$), regardless of the block we are in or which vehicle is selected, is always by design less likely to lead to success (percent success of reaching $g^+$, from data: 45% in high influence blocks, 3% in low influence blocks; of reaching $g^-$: 67% in high influence blocks, 43% in low influence blocks). Thus, the proportion of attempts on $g^+$ goals is a direct readout of the subjective trade-off between macroscopic controllability and reward, and will be a key outcome measure.

We report the proportion of $g^+$ attempts made using the more guidable vehicle ($v_1$), which we chose as a measure of sensitivity to microscopic control. This measure of course depends on the guidability inherent to $v_1$ (i.e. $\gamma(v_1)$). Finally, we report basic performance metrics such as the amount of money accrued and the number of losses, and examine how these are affected by control expectations as measured by LoC. Two trials in every block are catch trials (type *Catch* in the table in Fig 2); these see both vehicles placed at the center of the canvas and make it obvious that the choice should be of the most controllable vehicle (and pursuing $g^+$). We are interested in the proportion of correct choices here as a proxy for how important vehicular controllability was for a subject. In single goal trials, we just measure the trade-off between sensitivity to guidability and distance, via comparison of the choice proportion of the closer, less guidable (or equivalently of the farther, more controllable) vehicles.

## Model dependent analyses

Our model-dependent analyses were intended to provide a clearer view over the model-agnostic effects observed in raw data. We employed a hierarchical graphical structure for all our models [24]. All models contained various parameterised additive factors intended to estimate a net value of choosing a combination of vehicle and goal, given the features of the trial at hand. Posterior distributions over the parameters for each model for each participant and condition were estimated using a Hamiltonian Monte Carlo sampler, implemented in Stan (http://mc-stan.org; [25]). Model assessment was performed via the Widely-Applicable Information Criterion [26], and for completeness, also via leave-one-subject-out cross validation (LOSO), i.e. obtaining average log-likelihoods for each subject as they were held out of training and only used as test data. WAIC scores were used to perform model-recovery analyses on all our DGT models, the results of which are reported in S3 Fig [27]. We also generated synthetic data from our winning models to assess the fidelity with which they could recapitulate the statistical characteristics of the original behavior (S4 Fig), and to perform parameter recovery in DGT models (S5 Fig).

Statistical analyses related to LoC were based on permutation tests with empirical null distributions created by randomizing questionnaire scores across subjects. In addition to considering the relationship between parameters obtained and questionnaire scores, we tested whether the inferred parameter fits were *necessary* to recover the main model-agnostic effects observed in the raw data. For each parameter, this was done by (1) permuting its estimates across subjects, (2) generating synthetic data under the permutation (1000 data-sets), and (3) considering how many of the synthetic effect sizes, in the null distribution thus obtained, exhibited a larger magnitude than in the original labelling.

Finally, in order to establish the importance of recovered parameter estimates for *prediction* of individual LoC scores, we harnessed measures of conditional variable importance in the context of regression through random forests [28]. The rationale is the same as in the previous approach: a variable is shuffled and utilised in regression in a random forest, and the resulting mean squared error is compared with the value found with the original labelling.

The rationale underlying the use of these measures to establish (1) necessity for recovery of the model agnostic effect and (2) predictiveness of the parameters in relation to LoC-scores is that the former is helpful in understanding which parameter underpins the model-agnostic effect found, while the latter by-passes the choice of a model-agnostic outcome measure and provides a way of mapping the recovered parameters onto LoC.

The features of the trial that determined the additive components of the estimated value included vehicle-goal distance, vehicular guidability and reward size. These make up the value of choosing a vehicle-goal pair, which then generates decisions according to a softmax policy. Influence conditions were considered separately, with each including its own set of parameters. We added extra components to model behavioural change across the whole task.

For double goal trials, we incrementally built a sequence of increasingly more sophisticated models. We started from (1; "Additive") additive influences of vehicle-goal distance, vehicular guidability and reward size; and leading through (2; "Bias") a bias term indicating an absolute propensity to choose (or avoid) $g^+$, irrespective of the characteristics of the trial or any temporal aspect; (3; "Win-stay Lose-shift") a term which modelled the propensity to choose either vehicle to attain $g^+$, which varied as a function of the outcome of the previous trial, (4; "Vehicle independent RW") a Rescorla-Wagner term which promotes or discourages the choice of $g^+$ with either vehicle, by keeping track of a running average value of choosing $g^+$ (this conflates both vehicles into one value); and finally (5; "Vehicle dependent RW") a different Rescorla-Wagner term which does the same, but keeps separate records for the two vehicles. Note that, in DGTs, we do not consider interactions between predictors, as these would be too many to individually test.

For single goal trials, we assumed that subjects employed a policy based on (1) additive influences of vehicular guidability and vehicle-goal distance; (2) as in (1), but adding an interaction term of guidability and distance; (3) a combination of guidability, distance, and the pure probability of reaching the goal with either vehicle (the computation of which is shown in S2 Text); or, finally, (4) the pure probability. It is important to note that all these formulations amount to different approximations of the computation of probability; arguably, from the simplest, i.e. (1), to the most complete, (4).

Formal descriptions of the models along with their equations can be found in S1 Text.

## Results

### Model agnostic results

**Validity of control manipulations.**   We first assessed whether our control manipulations worked as planned. As intended, individual trial-averaged pressing frequencies were a strong predictor of performance in high influence blocks (money earned: $r = .71$, $p < 0.001$, 95%CIs [.50, .85]; num. of losses: $r = -.64$, $p < 0.001$, 95%CIs [-.80, -.40]). Conversely, due to the cap at $4Hz$, trial-averaged individual pressing frequencies were not significantly predictive of performance in low influence conditions (money earned: $r = .20$, $p = .24$, 95%CIs [-.14, .50]; num. of losses: $r = -.13$, $p = .44$, 95%CIs [-.44, .20]). Thus, by design, half of the blocks arranged a contingency between effort (in the form of pressing frequency) and reward, while the other half did not. Finally, as we hoped, vehicles' controllabilities could be discriminated quite reliably by the 16th, *Catch* trial (where both vehicles are placed at the center of the canvas), with a choice frequency of the more controllable vehicle averaging at 71% (high influence) and 73% (low influence), a good indication that by the time subjects were faced with SGTs, they generally knew which vehicle was most controllable.

**Double goal trials.**   We found that the proportion of $g^+$ attempts was anticorrelated with I-scores in both high ($r = -.50$, $p = .002$, 95%CIs [-.71, -.20]) and low influence conditions

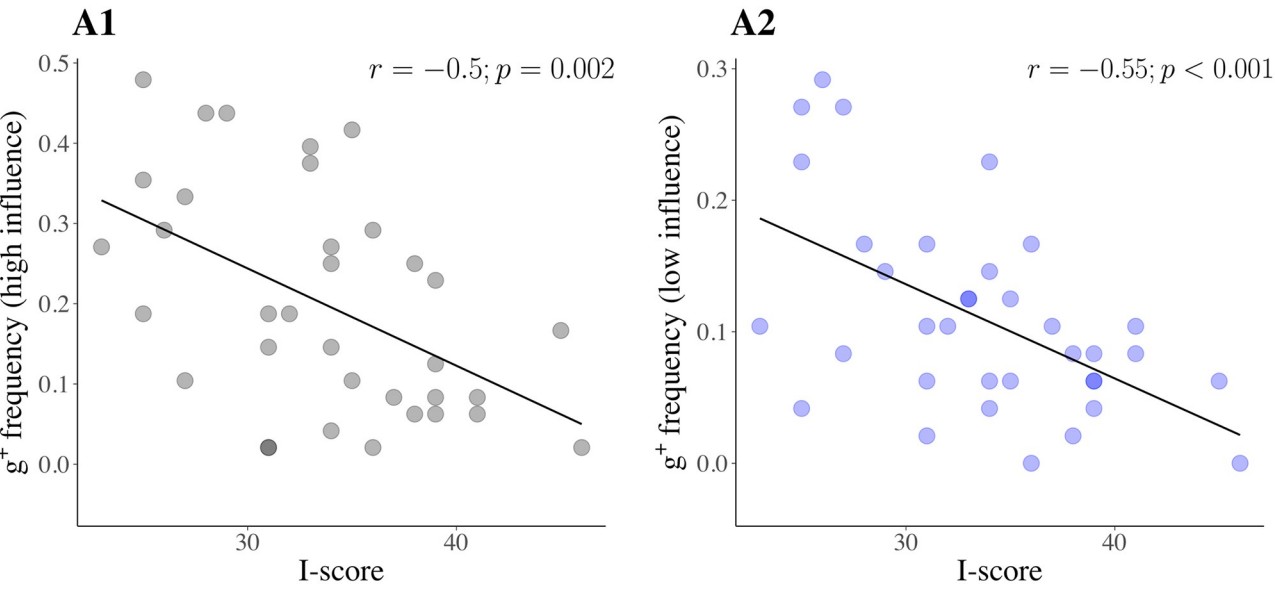

**Fig 5. The negative correlations between I-scores and choice frequency of $g^+$.** The figure shows the relationships found between I-scores and choice frequency of $g^+$ (the riskier, more rewarding) goals, in (**A1**) high influence and (**A2**) low influence conditions. Note the differing range of values (on the y-axes) which reflect the overall reduced choice of $g^+$, when influence over outcomes is diminished. These plots conflate all values of rewards associated with $g^+$, suggesting that the effect we observe is more to do with probability, rather than reward size.

($r = -.55$, $p < .001$, 95%CIs [-.75, -.27]) (see Fig 5); contrary to our predictions, we found no such relationship with respect to C-scores (high influence: $r = -.09$, $p = .61$, 95%CIs [-.41, .25]; low influence: $r = .20$, $p = .26$, 95%CIs [-.14, .49])). The vast majority of attempts to $g^+$ goals were made using the central vehicle (92%, in both influence conditions).

In high influence blocks, I-scores positively correlated with the probability of success, obtained via simulations, of the choices made ($r = 0.38$, $p = .02$ 95%CIs [.08, .72]), and inversely correlated with the average amount of money won when a win occurred ($r = -.37$, $p = .03$, 95%CIs [-.04, -.67]). Low I-scorers could sometimes attain $g^+$ goals (which they more often pursued), making up for their frequent losses with rare but conspicuous wins. Thus, the effects of probability and winnings balanced each other out so that we found no significant relationship between the amount of money earned and I-scores in high influence conditions. Conversely, in low influence blocks, we did not observe a correlation between I-scores and probability of success ($r = 0.19$, $p = .24$ 95%CIs [-0.15, .49]), or the correlation between the amount of money earned and I-scores ($r = -0.11$, $p = .43$ 95%CIs [-0.44, .20]); this is possibly because in low influence blocks the margin to make safer decisions is drastically reduced: all probabilities of success for all options are lower.

Although $g^+$ choice proportion was significantly reduced when vehicles were less guidable (permutation test $p < 0.001$; both influence conditions), I- and C-scores were not associated with preference for the more guidable vehicle when choosing to attempt $g^+$ ($p > .14$, I and C-scores; both influence conditions), or with the performance in discerning and choosing the most guidable vehicle on *Catch* trials ($p > .11$; I and C-scores, both influence conditions). This suggests, on a model-agnostic perspective (though this will be examined in more detail using modeling), that the microscopic action-to-outcome noise (or more abstractly, the objective quality of the tool used to carry out the plan) is not related to individual differences in LoC I- or C- scores.

Finally, $g^+$ choice proportion was strongly correlated across influence conditions ($r = 0.65$, $p <.001$, 95%CIs [.40, .81]), albeit being reduced when influence was low (permutation test $p < 0.001$). This suggests that the influence manipulation, and the drastic change in the overall achievability that follows (almost no chance of ever making $g^+$ using any vehicle), only caused a rescaling, not a flattening, of proportionate choice. I and C-scores were not associated with differential proportions of $g^+$ attempts across influence conditions, and so were not predictive of subjects' choices' sensitivity to the influence manipulation (I-scores: $r = -.26$, $p = .13$, 95% CIs [-.55, .07], C-scores: $r = .25$, $p = .13$, 95%CIs [-.08, .55]). Thus, perceived control is associated with behavioural signatures *within* each influence condition, but not with changes in strategy *across* the different influence conditions, despite the rather large difference in overall achievability between these.

**Single goal trials.**   I-scores correlated with choice of the closer vehicle in high influence blocks ($r = .39$, $p = .025$, 95%CIs [.06, .63]), but not significantly in low influence blocks ($r = .26$, $p = .12$, 95%CIs [-.07, .55]), regardless of the difference in guidability of the vehicles involved. No significant relationship was found for C-scores ($p > 0.43$ in either influence condition). I and C-scores did not predict rewards collected in SGTs, in either high influence (both $p >.58$), or low influence blocks (both $p > 0.31$); the latter is consistent with the fact that goals were only attained on a total of three occasions across all subjects and trials.

The frequencies of choices of the closer vehicle were strongly correlated across high- and low-influence blocks ($r = 0.46$, $p = .005$, 95%CIs [.15, .69]), reflecting similar decision-making strategies across influence conditions. We note that this result, in itself, constitutes evidence against a pure attainment probability-based class of models, as subjects had almost no chance of succeeding in low influence trials. As we shall see in our model-dependent analyses, this is also reflected in the poor performance of a pure probability-based model.

## Model-dependent analyses

We complemented our model-agnostic descriptions of the data with a fully-fledged computational picture that provides a richer analysis of the sensitivities to trial features (distance, vehicular controllability and reward size), and more critically, probes the learning mechanisms driving choices of the riskier $g^+$ goals. As anticipated in the Methods, we employed a hierarchical graphical structure for all our models, in which sensitivities to trial features interacted additively to produce the value of choosing a vehicle-goal pair. We assume that subjects' choices are algorithmically uniform across influence conditions (i.e., involve the same model components), but allow different sets of parameters for each condition. We generally report on our best computational accounts of both single and double goal trials.

**Double goal trials.**   In the best model, subjects learned about (the odds of achieving $g^+$ with) the two vehicles in a model-free way, throughout the task, using Rescorla-Wagner updates for their values, with separate learning rates for successes and failures. Vehicles then drive decisions in two ways: one that arises from the objective learning about their guidability (simply the update of a summary statistic of how many presses result in the correct movement), and the history of achievement which resulted from their use, in which successes and failures weigh differently, and in which the particulars of the problem at hand are absent.

The history of achievement of both vehicles is encapsulated in the terms $H_t(v_1)$ and $H_t(v_2)$. We show the evolution of the average of these terms (i.e. simply $H_t(v_1)/2 + H_t(v_2)/2$) through trials in Fig 6, for each influence condition (A1, gray dots: high influence; A2, blue dots: low infuence).

$H_t$ is an experience-derived proxy for the probability of reaching $g^+$ with a certain vehicle. This term (which takes values in the interval $[-1, 1]$; -1 meaning lowest, and +1 highest,

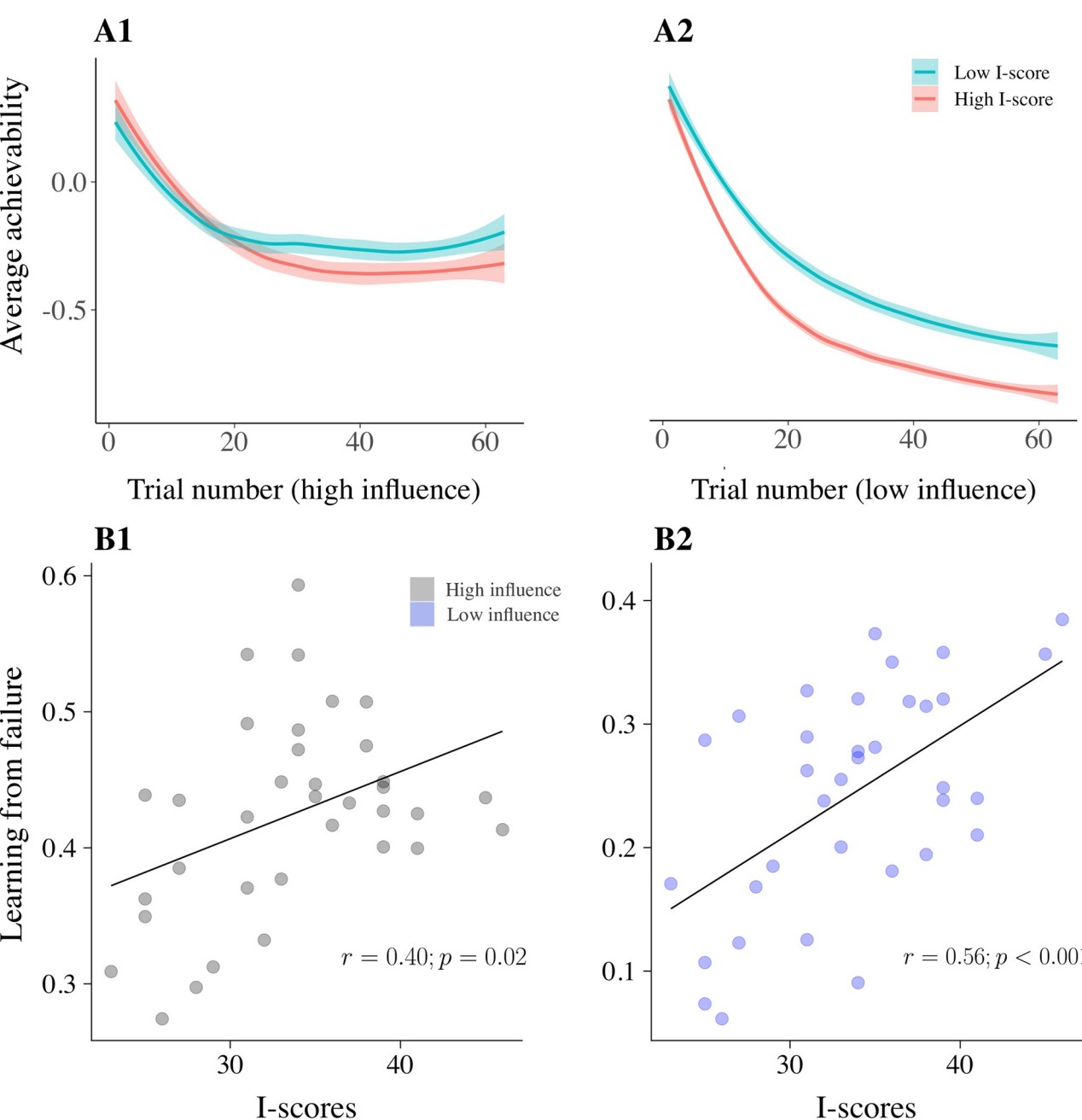

**Fig 6. Learning from failure and the evolution of achievability.** Plots **A1** and **A2** illustrate the evolution of average achievability (i.e. the parameter $H_t$) through DGTs in high and low influence conditions respectively. This is the *average* achievability because we consider the average of the values of $H_t$ for the two vehicles at each trial. The shaded error bands reflect standard error across subjects. The *x*-axis reports the ordered trials for each condition. This is a total of 64 trials (as there are 16, times 4, DGTs per each condition). For illustration purposes, we divided our subjects into two groups using a median split on I-scores. We can observe the average achievability learnt through Rescorla-Wagner updates decrease faster for high I-scorers in both influence conditions, however, the difference across the two groups is most apparent in low influence conditions. Plots **B1** and **B2** illustrate the correlations between I-scores and learning from failure in high and low influence conditions respectively, which underpin the temporal evolution of $H_t$ shown in A1 and A2. The effect is stronger in low influence conditions—however, note that the $\epsilon_l$ parameter in these conditions exhibits a measure of confusion, particularly with the initialisation (i.e. $H_1$) parameter (see S5 Fig).

perceived achievability) evolves through Rescorla-Wagner updates: it increases when a $g^+$ goal is achieved, and it decreases when *any* loss is incurred. For vehicle $v_1$ (the same exact updates apply, of course, to $v_2$), this will evolve as per the equation:

$$H_t(v_1) = H_{t-1}(v_1) + \begin{cases} 0 & \text{if achieved } g^- \text{using either } v_1 \text{ or } v_2 \\ \epsilon_w(+1 - H_{t-1}(v_1)) & \text{if achieved } g^+ \text{using } v_1 \text{ at } t-1 \\ \epsilon_l(-1 - H_{t-1}(v_1)) & \text{if lost using } v_1 \text{ at } t-1 \end{cases} \quad (1)$$

Here, $t \in \{1...16\}$ indicates the current trial. When $t = 1$ (first trial of the task), $H_1$ is equal to the prior value for achievability. $\epsilon_w$ ($\epsilon_l$) are the learning rates for success (failure). Note that this update only takes place for the vehicle that was chosen at time $t - 1$, in this case $v_1$. The value of $H_t$ for $v_2$ remains the same, i.e. $H_t(v_2) = H_{t-1}(v_2)$. The rationale of this learning rule is that learning by experience in this task should take into account that (1) failure to achieve *any* goal should imply that the odds of achieving $g^+$ should be lower (as $g^+$ is always harder); (2) success in attaining $g^-$ should not rationally imply that $g^+$ is more achievable (since $g^-$ goals are always easier) (3) achieving $g^+$ is the only evidence that counts towards achievability of $g^+$ with a certain vehicle. For completeness, we tested a further model which had a separate learning rate for achievements of $g^-$ goals, which performed less competently. Further strengthening our argument for "Vehicle dependent RW" being a competent model (i.e. that subjects used reward-history to learn vehicle value) is that the inclusion of the $H_{16}(v)$ term (i.e. the vehicle value on the last—16th—trial) improved WAIC scores in SGT models—so that the history of (un)achievement of a vehicle carries over to SGTs, and plays a role in decision-making. In Table 2, we report LOSO and WAIC scores for all our models. In S4 Fig, we show the general calibration plots for choice frequency of the more rewarding goals ($g^+$) and more controllable vehicles ($v_1$), demonstrating the generative adequacy of the winning model.

Post-hoc correlations of the parameters from the winning model revealed that the main determiners of the relationship between reduced choice of $g^+$ and I-scores were the increased learning rates (learning from failure) of the model-free term (in both high, and low, influence conditions) and increased distance sensitivities (in high influence conditions). In low influence conditions, we registered lower initialisation values (i.e. $H_1$) in high I-scorers.

Reward sensitivities anti-correlated significantly with I-scores, but they were not essential for the recovery of the main effect of reduced $g^+$ choice when compared to other variables (see Fig 7).

Further, in order to obtain a global view of variable importance (not towards the recovery of the model agnostic effect concerning choice frequency of $g^+$, but towards prediction of I-scores) we used random forests ('party' package in R) to predict I-scores using all subject-wise parameter estimates. Here, variable importance was measured as the mean increase in the

**Table 2. Model comparison for DGTs.** We report out of sample subject-wise likelihoods (LOSO scores) and WAIC scores for all our models (lower scores are better). Starting from the easiest (additive), which forms the basis for all our models, to the most complex (and best performing). LOSO and WAIC only disagree over models 3 and 4, with WAIC preferring number 4.

| Models | LOSO | WAIC |
|---|---|---|
| Additive (1) | 0.623 | 5213 |
| $g^+$ bias (2) | 0.632 | 5180 |
| Win-stay Lose-shift (3) | 0.644 | 5179 |
| Vehicle independent RW (4) | 0.637 | 5143 |
| Vehicle dependent RW (5) | 0.653 | 5118 |

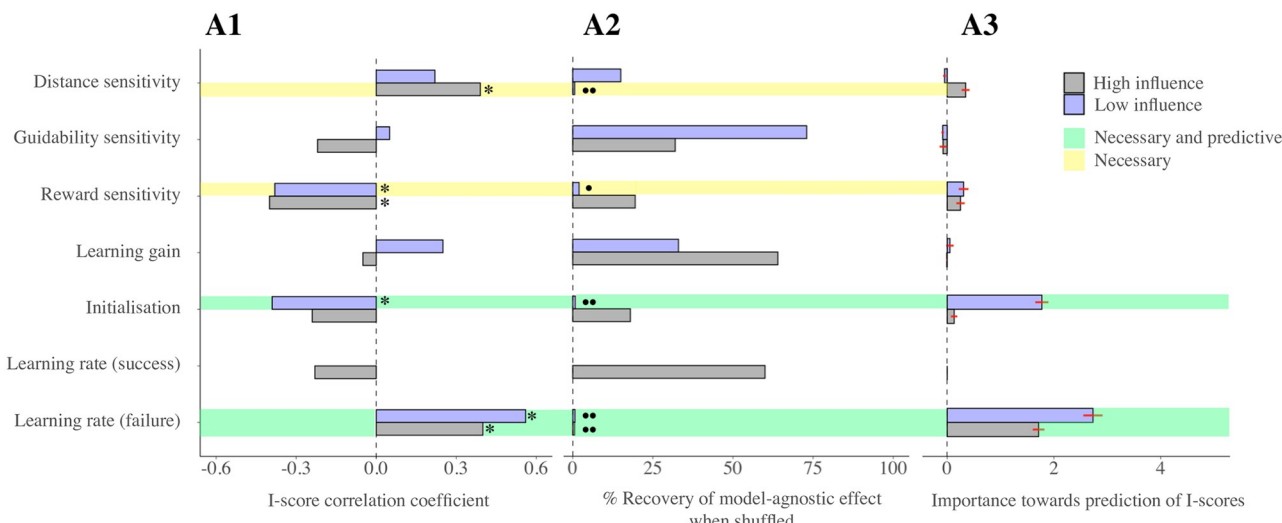

**Fig 7. I-scores and parameter estimates.** This series of plots illustrates the relationship between I-scores and the recovered parameter estimates of the winning model (i.e. 'Vehicle dep. R-W', in Table 2) in three ways. Note that learning rate from success (in attaining $g^+$) is absent in low influence conditions, as only 3 wins over all datasets were recorded. **A1** illustrates the correlation coefficients between each parameter and I-scores. Next to each bar, a star (*) indicates significant correlations post false discovery rate correction for multiple comparisons. In **A2**, each bar exemplifies the percentage of recovery of an equal or larger correlation coefficient of I-scores and $g^+$ choice frequency (the effect found in model-agnostic analyses) when shuffling the recovered parameters across subjects. We performed this analysis (described in Methods) to understand which parameters played the most important role in driving this effect. Two circles next to the bar indicate a percentage of recovery smaller than 1%, one a percentage smaller than 5%. Thus, according to this measure, the effect is driven by distance sensitivities (the most apparent feature of achievability) in high influence conditions, and (in both influence conditions) by the learning rates from failure. Finally, in **A3**, each bar shows the loss in predictive power of I-scores when using the recovered parameter estimates as predictors. This measure used random forests, and was obtained as described in Methods. The errorbars signify the standard errors (the procedure was repeated 50 times). Finally, as the legend shows, the parameters highlighted in green are those which show significant effects with I-scores, are needed to reproduce the model agnostic effect, and exhibit a very high importance for I-score prediction. In yellow, we highlighted those parameters which, albeit being necessary to reproduce the model-agnostic effect, turned out not to be critical for predicting I-scores.

squared residuals of predictions (estimated with out-of-bag cross-validation) as a result of mean parameter estimates being randomly shuffled. The results were consistent with what we found for the recovery of the main model-agnostic effect, but placed an even stronger emphasis on the importance of learning rates because of their relevance to the prediction of I-scores (see Fig 7). Note however that the lower initialisation values in low influence conditions are also prominent predictors of I-scores. Consistent with this observation, our parameter recovery analyses found a measure of confusion between parameters $H_1$ and $\epsilon_l$ in low influence conditions (S5 Fig). This is likely the reason that both are involved in predicting I-scores, and stems from the fact that learning of unachievability happens very quickly as influence is so greatly compromised. This speed of learning means that there are too few trials to disentangle biased initialisation from fast learning. In high influence conditions, $\epsilon_l$ is moderately well recovered despite strong posterior correlations that arise with the learning gain parameter (i.e. $\alpha_H$; see S5 Fig).

Finally, after having established that learning models performed better (i.e. models 3,4, and 5), we went on to test a further class of models in which subjects learned about the values of (subsets of) task-relevant features (i.e. distance, guidability and reward) simultaneously. We report details of the results on this approach in S3 Text. For now, we just note that none of these models achieved a better performance than "Vehicle dependent RW", though some performed quite well.

**Table 3. Model comparison for SGTs.** The most parsimonious model accounts for data through adding distance and vehicular guidability features, and their multiplicative interaction. I-scores were explicative of distance sensitivities, but only in high influence blocks.

| Model | Description | LOSO | WAIC |
|---|---|---|---|
| 1 | Additive | 0.554 | 1280 |
| 2 | Interaction | 0.584 | 1273 |
| 3 | Additive + Probability | 0.543 | 1286 |
| 4 | Probability | 0.521 | 1488 |

**Single goal trials.** In Table 3, we report results for all our models for SGTs. Note that an extra term is added for all models which embodies the reward history-dependent value of a certain vehicle as recorded by the 16th double-goal trial (i.e. $H_{16}(v)$) as addition of this term was found to improve model performance (see S1 Text, paragraph "Model-dependent analyses: single-goal trials" for the exact formulation). We found that a model based on the simple additive interaction of vehicular guidability (which, by the time of the SGTs, we assume to be known), vehicle-goal distance, and their interaction, yielded the most parsimonious account of the data. Consistent with our model agnostic findings, we found no significant relationship for C-scores, while I-scores correlated with distance sensitivity in high influence blocks ($r = .40$, $p = .01$, 95%CIs [0.08,0.65]). Internals then appealed to a more apparent cue of achievability (the vehicle-goal distance) in order to maximise the probability of success in SGTs; however, this did not quite work out in the same way as with DGTs, as here I-scorers did not make more probably successful decisions on average. By design, in fact, it is best to choose the central vehicle in two out of the four SGTs at the end of each block (this is described in Fig 3).

Importantly, the focus on distance in SGTs reflects the value attributed to this feature during DGTs; distance sensitivity parameters were in fact correlated across S and DGTs (high influence: $r = 0.49$, $p = 0.002$, 95%CIs [.19, .70]; low influence: $r = 0.65$, $p < 0.001$, 95%CIs [.35, .78]).

Perhaps due to the near flat probability of reaching the goal inherent of low influence conditions SGTs, neither distance nor vehicular control sensitivities correlated with I-scores (distance s.ty: $r = 0.21$, $p = 0.22$, 95%CIs [-.13, .5]; veh. guidability s.ty: $r = -0.30$, $p = 0.08$, 95%CIs [-.57, .03]).

## Exploratory analyses

**Powerful others.** We did not have preliminary hypotheses regarding the P-scale. However, additional analyses of our data revealed that money earned in low influence DGTs (but not SGTs, recall these almost always led to losses by design) was very strongly positively correlated with the P scale $r = .57$, $p < 0.001$, 95%CIs [.3, .76]); P-scorers' choices in low influence trials were also less likely to lead to a loss, whichever the amount ($r = -.49$, $p = 0.003$, 95%CIs [.18, .7]). see Fig 8. We investigated which parameters could be responsible, rather as we did for I-scores, and found that the distance sensitivities parameters correlated heavily with P-scores ($r = .48$, $p = 0.003$, 95%CIs [.17, .7]) and were necessary to recover the effect of increased earnings ($p = 0.02$) and choice probability.

## Discussion

We designed a novel decision-making task which requires subjects to learn and adapt to the controllability of environments, and thereby allows us to isolate distinguishable components relevant to control. We analyzed the results using both model-agnostic assessments of choice

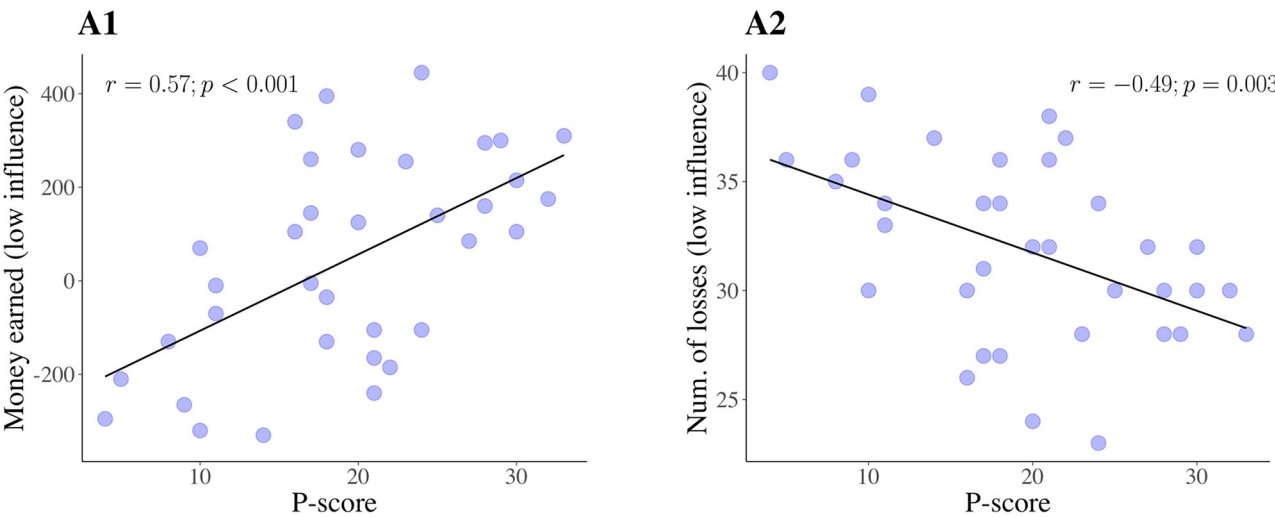

**Fig 8. P-scores and performance in low influence conditions.** Plots illustrating (**A1**) the positive correlation between P-scores and money earned, and (**A2**) the negative correlation between P-scores and number of losses, in low influence trials.

frequencies and model-dependent characterizations of the trial-by-trial effects of rewards and punishments.

We found correlations between specific locus of control (LoC) scores and both model-agnostic and model-dependent aspects of performance and learning. Internal (I)-scores correlated with a form of "timidity" in the shape of reduced choice of the less attainable, more rewarding ($g^+$) goals. This correlation was significant when subjects had either greater or lesser influence over the environment. In the high influence condition, the timidity of high I-scorers entailed that they had a higher likelihood of success (albeit not gaining more points on average, because this meant by design that they chose the impoverished goals).

Our computational analyses favoured a model in which value-based decision making was based on a form of Rescorla-Wagner learning of how achievable the more attractive goal ($g^+$) was with each vehicle (a term we call $H_t$). There was evidence for separate learning rates for appetitive (success) and aversive outcomes (failure) [29–34], with the latter being higher, as has previously been observed in other circumstances [32, 35].

We then showed that the boosted learning from failure and biased initialisation of achievability ($H_1$) were instruments of the negative relationship between $g^+$ goal choice frequency and I-scores observed in our model-agnostic analyses. Crucially, our random forests-based analyses established that these were substantially the best predictors of internality.

Our observations are broadly consistent with existing observations about the role of LoC in decision making, including the fact that internal subjects tend to place more conservative bets [19], opt for courses of action allowing lower chances to incur in episodes of failure [20], and expend more effort in a task when its goal is framed as avoiding punishment, rather than attaining reward [36]. However, our findings broaden these results through our focus on learning.

Learning was also the focus of Rotter and his group, who utilised explicit *verbal* manipulations of the contingency between actions and outcomes to study the evolution of explicit expectations [5]. Different subject groups would typically be provided with different sets of instructions, with the same task framed as a matter of "Skill", or "Chance" (there would sometimes also be "ambiguous" instructions). The impact of reinforcement history on expectations over future outcomes was then probed.

Changes in expectation were blunted in "Chance" conditions [5, 16–18]. Under "Skill" conditions, subjects quickly increased their expectations of success after wins, and only slowly learned their inability to succeed after failure [18]. The discrepancy between this result and our observation of faster learning from failure might arise from a difference between control (as in I- scores) and the spectrum between skill and chance (as in Rotter's work).

That we found the LoC 'Chance' C-score to be unrelated to any aspect of performance suggests that the task was largely perceived as a matter of skill. This, in turn, makes it very unlikely that the learning effect we observe is due to a form of gambler's fallacy (i.e., an expectation of achieving $g^+$ now because it has not been achieved in a while), as this latter is more of a feat of high beliefs in Chance [5].

By contrast, the strong link between the LoC 'Powerful-Other' P-scores and performance in low influence conditions was unexpected, and so merits replication. One possibility is that subjects with high P-scores might be more fatalistically adaptive to circumstances in which a source of control exists, but is beyond reach. It is worth noting that the experimenter was present throughout the experiment, possibly increasing discomfort for those with low P-scores, and affecting performance.

The tie we observed between high I-scores and boosted learning from failure is relevant to converging evidence from multiple different disciplines about the role of perceived controllability in coping in the face of negative events [37–39], or stress [40, 41]. Perceived control regulates emotional and behavioural responses to aversive outcomes, both when availing of situational manipulations (see, e.g. [2]), or using LoC questionnaires to measure generalised expectations [37, 42, 43]. For instance, perceiving an aversive stimulus as controllable modulates the neural responses associated with its presentation [44] or its predictive cue [45], and reduces anticipatory anxiety [46]. The picture that emerges from combining this literature with our results is one where dysfunctional reactions to perceived unachievability (excessive frustration, stress, or anticipatory anxiety; such as those found in the second experiment in [20]) might underpin a blunted capacity to learn from (and adapt to) it. To make matters worse, this mechanism easily self-sustains: failure without adaptation entails more exposure to failure, and is bound to reinforce an idea of low internality. From a clinical perspective, our results point to the successful integration of negative information about the self (failure) as a prominent factor for internality, and point towards a computational trait (learning from failure) which has the virtue of guarding subjects from loss, and subserves the maintenance of a healthy idea of controllability, and might be lacking or weak in many disorders. Our observations are then relevant to those disorders in which the integration of aversive information would be a requirement for alleviating symptoms—such as those characterised by perseverative dysfunctional thoughts or behaviours. However, it is only through further applications of our task in clinical populations that we will gain more insights into these possibilities.

In a study involving LoC, it was found that learning-related changes in the emotional response to negative outcomes were mediated by activity in ventromedial prefrontal cortex (vmPFC); and that, as individuals moved towards the external end of the LoC spectrum, the vmPFC response to predictable threat decreased (the study, however, could not differentiate between internality, chance and powerful others) [43]. The vmPFC has duly been implicated in computations involving control detection [43–48], and, in studies in rats, has been shown to suppress the over-exuberant activity of serotonergic neurons in the dorsal raphe when they have control over aversive outcomes [49]. The latter is one of the sources of evidence that serotonin is involved in aversive learning [50–52]. Our study was not designed to reveal the neural substrates of the decision-making strategies involved, and in particular the excess learning

from failure. From the original framing of the task as involving controllability, the vmPFC would be an obvious target of investigation, although one should note that, under our operationalization of this term, it had only a relatively modest impact on the behavior.

Despite these promising findings, we should note some caveats which concern our results. For instance, while we found no significant relationships involving the Chance sub-scale, it is entirely possible that these might exist, but simply play less apparent roles than the I- (or P-) sub-scales. Replications of our study, possibly availing of larger sample sizes, could offer further support to our results. From the perspective of determining individual differences, it would be desirable to increase the fidelity with which we can recover parameters (from simulated data). This might be achieved by presenting a single ('intermediate') influence condition, rather than two (since it was difficult, for instance, to disentangle low initialisation values from fast learning in low influence conditions). In this case, a single set of parameters would be constrained by all 160 trials. From a design perspective, it would be desirable to have a range of different losses, so that sensitivity to negative outcomes could be assessed; it would also be important to present the threat of primary aversive outcomes such as shocks. Further, we implemented control in a rather direct manner, using effort. It would be interesting to see if our findings generalized to a simpler choice task. Finally, the task lacked the sort of complexity that would allow a distinction between different mechanisms for choice such as model-based and model-free learning and planning [53]. Controllability, and indeed biased learning from success and failure, might affect each differently.

In sum, we have provided a new tool for investigating controllability, and new results about the link between internality and biased aspects of learning. Our findings are consistent with the proposition that internality is subservient to giving negative information more weight, in a way that facilitates adaptation, or as Lefcourt would put it, more positively [37].

## Supporting information

**S1 Fig. Instance of a block from trials 1 to 16 (DGTs).** The background of the canvas is light blue, indicating the influence condition (for half of the subjects this would signify high, and for the other half, low, influence). The vehicles and goals for the first 16 trials of a block are displayed as they appear at the start of the trial (vehicle selection phase; the red timer on top denotes time left to choose, random here) with goals made slightly bigger, for illustrative purposes. The ordering of the trials is randomized, as is the position of the goals (the angle by which they are rotated). The labels for each trial type are on top of each screenshot, the number in parentheses being the order of occurrence of this trial type within the block. Note how on the second occurrence (i.e. (2)), the vehicles' position is swapped. The sixteenth trial (bottom-right) is *always* a catch trial; but the ordering of all the other trials is random.
(TIF)

**S2 Fig. Winning model ("Vehicle dependent RW"): recovered parameter distributions.** Here, we show the recovered parameter fits for our winning model, obtained via stan. Learning from success ($\epsilon_w$) is inherently only defined in high influence conditions. All values are on average lower in low influence conditions, possibly due to the greater stochasticity in decision making (this is equivalent to having a lower inverse temperature). $H_1$ values in low influence conditions are slightly higher, possibly compensating for the incapability of the model to describe higher, initial tendencies to choose the riskier goal through other trial-based features. Although learning from failure ($\epsilon_l$) parameter fits are lower on average in low compared to high influence conditions, recall that this is the only learning feedback we posit for low influence conditions. Thus, the resulting $H_t$ values will, in time, quickly fall below those reached in

high influence conditions, as they can only monotonically decrease.
(TIF)

**S3 Fig. Confusion matrix for model recovery.** We generated 50 datasets from all our models, and performed model recovery analyses on all models (Palminteri et al. 2017), using WAIC scores. We generated synthetic data from the posterior parameterisation of each model obtained after fitting to the data. Similar models are at higher risk of being confused—however, our winning model ("Vehicle dependent RW") had an expected probability of 0.04 of being wrongfully recovered if a different formulation were true (false positive rate); and a probability of 1 of being recovered if the data were truly generated by it (true positive rate).
(TIF)

**S4 Fig. Calibration of goal and vehicle choice frequencies.** These plots show that the winning model ("Vehicle dep. RW") generated frequencies of $g^+$ choices (left) and more controllable vehicle $v_1$ (right) are well calibrated to the data. Each dot is a subject's statistic as computed in a particular influence condition (gray: high; blue: low). The variance explained by the model is high for both measures ($g^+$ and $v_1$ choice frequency) and influence conditions (high and low): (i) $g^+$: high influence, $r^2 = 0.96$, low: $r^2 = 0.84$, (ii) $v_1$: high influence, $r^2 = 0.69$, low: $r^2 = 0.49$. Low influence statistics are less faithfully recapitulated than high influence statistics, possibly on account of subjects inherently attempting to randomise their decision making to obtain more success. Finally, vehicular choice is less faithfully recapitulated in both influence conditions—this is expected, since subjects only have a noisy notion of which vehicle is most controllable.
(TIF)

**S5 Fig. Parameter recovery in DGTs.** Here, we show recovery analyses results for the winning model parameters (i.e. sensitivities to distance, $\alpha_d$; to reward, $\alpha_r$; to guidability, $\alpha_v$; learning gain, $\alpha_H$; learning rates from success, $\epsilon_w$; from failure, $\epsilon_l$; and the initialisation, $H_1$). The confusion matrices exemplify the intrinsic correlations in the posterior parameter means (**A**; high influence and **B**; low influence conditions) and the capacity of our fitting procedure to to recover the original parameters when these generate synthetic data (150 datasets; **C**, high influence; **D**, low influence conditions). For the latter two insets, we chose to report the variance explained or $r^2$ as a clearer measure of the amount of information about the original parameters that is carried by the recovered parameters. Parameters $\epsilon_l$ and $\alpha_H$ exhibit rather large correlations in the posterior (inset A). However, recovery is good in high influence conditions, as the variance explained for all parameters is maximal for the corresponding parameter (inset **C**). In particular with respect to our results and conclusions, recovery of the $\epsilon_l$ parameter is sufficiently good in high influence conditions. In low influence conditions, we find a measure of confusion between parameters $H_1$ and $\epsilon_l$, which are respectively the initialisation, and the learning (from failure) components of the Rescorla-Wagner component of the model. Thus, functional interpretations of individual differences here should be interpreted with caution.
(TIF)

**S1 Text. Model-dependent analyses.**
(PDF)

**S2 Text. Computation of probabilities.**
(PDF)

**S3 Text. Feature-based learning models.**
(PDF)

## Acknowledgments

We would like to thank Elena Zamfir, Tore Erdmann, and Chris Mathys for interesting discussions and comments.

## Author Contributions

**Conceptualization:** Federico Mancinelli, Jonathan Roiser, Peter Dayan.

**Data curation:** Federico Mancinelli.

**Formal analysis:** Federico Mancinelli, Peter Dayan.

**Funding acquisition:** Peter Dayan.

**Investigation:** Federico Mancinelli, Peter Dayan.

**Methodology:** Federico Mancinelli, Jonathan Roiser, Peter Dayan.

**Project administration:** Peter Dayan.

**Software:** Federico Mancinelli.

**Supervision:** Jonathan Roiser, Peter Dayan.

**Validation:** Federico Mancinelli.

**Visualization:** Federico Mancinelli.

**Writing – original draft:** Federico Mancinelli, Peter Dayan.

**Writing – review & editing:** Federico Mancinelli, Jonathan Roiser, Peter Dayan.

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
