## [Decision Letter · Decision Letter 0]

27 Aug 2020

Dear Prof. Dayan,

Thank you very much for submitting your manuscript "Subjective Beliefs In, Out, and About Control: A Quantitative Analysis" for consideration at PLOS Computational Biology.

As with all papers reviewed by the journal, your manuscript was reviewed by members of the editorial board and by several independent reviewers. In light of the reviews (below this email), we would like to invite the resubmission of a significantly-revised version that takes into account the reviewers' comments.

Dear Federico, Jonathan and Peter,

Thank you very much for submitting your manuscript to PLOS Computational Biology.

This submission was chosen as part of a new pilot project on Reproducibility of models by PLOS CB. Therefore, in addition to the standard review process, reviewer # 1 from the Center for Reproducible Biomedical Modeling evaluated the possibility of reproducing the result based on the current status of submission only. For more details of this pilot, you can refer to these links: https://biologue.plos.org/2020/05/05/improving-reproducibility-of-computational-models/

https://journals.plos.org/ploscompbiol/article?id=10.1371/journal.pcbi.1007881

This reproducible review and another review both pointed out the lack of availability of data and modeling code, which makes it difficult for other researchers to reproduce the result. This should be addressed in addition to other comments.

The reviewers have raised important issues that I believe need to be carefully addressed. The worry about statistical power, the lack of clarity in the current way of writing, and the rationale of choosing the questionnaire are common issues raised by the reviewer. In addition to the raised issues, it is worth improving the introduction and discussion such that a clearer take-home message can be delivered. A clearer connection to the existing research and theory should also help more readers to understand the value of your work.

I hope these reviews can help you improve the manuscript and I look forward to seeing your updated version!

We cannot make any decision about publication until we have seen the revised manuscript and your response to the reviewers' comments. Your revised manuscript is also likely to be sent to reviewers for further evaluation.

Sincerely,

Ming Bo Cai

Associate Editor

PLOS Computational Biology

Daniele Marinazzo

Deputy Editor

PLOS Computational Biology

Dear Federico, Jonathan and Peter,

Thank you very much for submitting your manuscript to PLOS Computational Biology.

This submission was chosen as part of a new pilot project on Reproducibility of models by PLOS CB. Therefore, in addition to the standard review process, reviewer # 1 from the Center for Reproducible Biomedical Modeling evaluated the possibility of reproducing the result based on the current status of submission only. For more details of this pilot, you can refer to these links: https://biologue.plos.org/2020/05/05/improving-reproducibility-of-computational-models/

https://journals.plos.org/ploscompbiol/article?id=10.1371/journal.pcbi.1007881

This reproducible review and another review both pointed out the lack of availability of data and modeling code, which makes it difficult for other researchers to reproduce the result. This should be addressed in addition to other comments.

The reviewers have raised important issues that I believe need to be carefully addressed. The worry about statistical power, the lack of clarity in the current way of writing, and the rationale of choosing the questionnaire are common issues raised by the reviewer. In addition to the raised issues, it is worth improving the introduction and discussion such that a clearer take-home message can be delivered. A clearer connection to the existing research and theory should also help more readers to understand the value of your work.

I hope these reviews can help you improve the manuscript and I look forward to seeing your updated version!

Reviewer's Responses to Questions

**Comments to the Authors:**

Reviewer #1: Reproducibility report has been uploaded as an attachment.

Reviewer #2: Review of Mancinelli et al

The authors develop a novel task to measure how people learn action-outcome controllability when making goal-directed decisions, and how these judgements are related to beliefs about locus of control (LoC). They found that participants incorporated a model-free estimate of the value of different agents (cars they would drive towards different goals), and that it was the learning of these agents that was most strongly associated with ‘internalizing’ factors of LoC. A major finding was that high LoC allowed subjects to quickly learn when a task was unachievable, so that they did not incur further loss and affording an attribution of failure to external rather than internal causes.

Overall, it’s an interesting task, analysis, and interpretation but there are some substantial concerns that need to be addressed.

Major Concerns

1- More could be done to ensure that the RW components reflect the learning of vehicle value, and not learning the properties of each vehicle. This concern arises because the reward information that’s used to teach the vehicle value depends in part on the vehicle properties (esp. guidability). When people get a bad outcome, how do we know that they blame the vehicle as a whole, and not blame a property of the vehicle. A deeper investigation into this learning process may help inform the surprising relationship between internalizing and learning.

a. The authors should determine whether participants use reward feedback to update their weights on different properties (influence, guidability, distance), in order to better distinguish where participants assign credit for bad performance.

b. It would be helpful to better understand what your estimate of the RW terms would look like under different generative models (and to validate the discriminability between models with model recovery).

c. Was there any evidence that the vehicle value biased decision in the single-goal phase?

2- Just based on this survey, it’s hard to make strong claims about internalizing per se, relative to other traits that may be associated with internalizing. A related concern is that this sample size is small for studying individual differences, so influences that one might expect to play a larger role (like mood, SES, education, IQ, OCEAN) may be more likely than a specific LoC subscale.

a. The authors should assess the reliability & factor structure for the internalizing scale (and the eigenvalues of the subscale covariance matrix would be more informative than the pairwise correlations).

b. The authors should avoid over-interpreting the implications of neuroimaging experiments on LoC based on these results.

c. Ideally a replication (e.g. online) would increase confidence in the conclusions and allow other factors to be disentangled.

3- It would be great if the authors would formalize this decision process in terms of the optimal choice strategy.

a. How good are the different vehicle property weighting strategies (additive, multiplicative, RW, etc) at predicting the reward probability?

b. The conceptual differences between influence and guidability are interesting, and could be further elaborated. How should choice algorithms use these two sources of information (e.g,. How would they alter the transition matrix of a model-based controller)?

Minor Concerns

The title is uninformative - would be more helpful if it summarized the main finding.

Reviewer #3: In this paper, the authors have designed a new task in which participants choose between different tools (cars) to reach various simple goals (targets leading to rewards - punishment if they don’t reach any goal). The cars are only partially reliable (noisy) and intermittently effective (the effort exerted by the participant has more or less influence) to degrees that need to be learned. The authors are interested in the relationship between participants’ scores related to locus of control/ internality with respects to self (Levenson I-scores) and their performance at the task (in particular whether participants choose the more rewarding goal which is taken as a read out of their subjective tradeoff between achievability and reward). Computational analyses showed that the negative relationship between the most rewarding goal choice frequency and I-scores was explained by boosted learning from failure and biased initialisation of achievability.

I found this paper, and more generally the question regarding subjective perception of controllability, very interesting and the methodological aspects are novel, original and solid. The paper will be inspiring for the computational psychiatry community.

However, I think the paper (and abstract) could be reworked to make it a clearer and more impactful read.

The questions asked, their significance, the hypotheses and important of the results could come across in a clearer way. Similarly the logic behind the design of the task could be clearer, e.g. it is initially unclear whether the authors are interested mostly in influence of initial expectations about control, aspects of learning or decision-making, what the hypotheses are with respect to the questionnaires (what patterns of behaviour would be expected under different hypotheses and are important to distinguish) and how they could generalise to more meaningful situations. The relevance and application of the results to mental illness could also be clarified (e.g. are the questionnaire scores relevant to dimensions such as low mood? anxiety? do the results have implications for psychotherapy?).

Regarding the results, it is also unclear initially whether the results could simply be interpreted in terms of high-I participants preferring more controllable choices (after having learned which are controllable - which would sound like a somewhat trivial result).

I was also wondering how the authors’ results could be interpreted in terms of / are controlled for risk and loss aversion (and possibly anxiety) and what was the rationale for not using potentially relevant different questionnaires in addition to locus of control.

Some aspects are confusing for e.g. in the abstract and some places in the text “High internality (Levenson I-score) was associated with decision making patterns that are less vulnerable to failure, and at the same time less oriented to more rewarding achievements.” If I understand correctly, the latter result (less oriented to more rewarding achievements) is highly dependent on the design of the task (where choosing the more rewarding goal is by design less likely to lead to success). It seems to me that it would be important to distinguish results that are general and potentially meaningful in wider contexts, from those that are specific to the task design, otherwise the take home messages become unclear.

- In Author summary: “Our findings can be interpreted within a theoretical framework which

implicates control expectations in individual learning differences, in a way that resonates with perseveration and also fit well within modern theories of learning in aversive contexts and serotonergic function. “ In absence of further explanation, I found that sentence unclear.

Reviewer #4: This is a well written manuscript on an important matter. I read this manuscript with great interest and I very much appreciate the task design and efforts with the computational procedures. I have some major and minor comments for the authors, they should consider for revising this manuscript.

Comment #1: While the task design for itself is clever, I am not sure whether the authors have sufficiently embedded the task in the existing body of literature and theory on the concept of control. This is rather vaguely justified in the introduction with the statement “Thus, we lack a task that decomposes objective controllability into its finer parts described above, and assesses the interaction between outcomes, ascription style, expectations and choices.” What exactly are the finer parts described above? At least for me, this is unclear and more importantly I cannot find a clear description on how these map on the current task design and the specific model parameters that are used to explain behavior. I think the specifics of the task design need to be much more thoroughly and clearly integrated to the existing literature so one can understand the rationale for why this task is needed and useful to understand an important aspect of controllability. More precisely, which aspects of the parts of controllability do the manipulations of “influence” or “guidability/reliability/contingency” address that have been brought up as relevant in the literature? Or are these just necessities for examining learning? Why are the different "influence" conditions needed? I very much believe that this task merits a more profound introduction and justification here.

In this line, the summary from l 45 on is somewhat difficult to understand based on the brief description. The “reliability” at the beginning refers to the “contingency” later but I would recommend introducing one concept – as has been done with influence - and keep refereeing to it in the following. Later in the manuscript you use guidability. I am honestly not sure which concept best differentiates the two components of your design, but I would introduce something here that you explain and stick to.

Comment #2:

I fear that this study might not be sufficiently powered, or that the expectations the authors are having with regards to the underlying effect size is too optimistic. That is a limitation of the study, that should be discussed. .45 is quite a large effect for correlational measures and this study is even underpowered for medium effects. On which ground do you assume that the associations you are looking for might be this large?

On a related note, I was wondering if the authors have considered that they always tested two scales and corrected for the associated increased Type 1 error.

Comment #3:

ll278 – 285 (i) Would you be so kind a provide the report of the results identical for the high and low influence blocks. Your report for the high-influence block is more complete compared to the low influence block without further explanation. Also, if you find no significant relationship between the amount of money earned and I-scores, please provide the relevant statistics for this. (ii) Related to this, I was wondering if the associations significantly differed between the two influence conditions and whether this would be important for your set of hypotheses to test? If you want to discuss findings as if they differ between the conditions, you should also test it here, if I understand correctly (as you e.g. do in l 404). Please watch carefully, that if you want to discuss associations to differ between conditions, as being smaller or stronger, that you also test the difference, because otherwise this claim is not backed up by statistical inference.

Comment #4:

I am not sure if I would agree to your evaluation of the results in ll 292-294: “This suggests, on a model-agnostic perspective (though this will be examined in more detail using modeling), that the microscopic action-to-outcome noise (or more abstractly, the objective quality of the tool used to carry out the plan) is not a particularly prominent feature of controllability in this task.”

As far as I understand it is just not significantly related to individual differences in the two facets of controllability, but that does not imply it is not a prominent feature of the task. It might be extremely relevant so that anybody, whether experiencing high or low control in everyday live, would pick the tool with better quality. I would suggest rephrasing this section.

Comment #5:

Could you provide internal consistency measures for the LOC scales?

Comment #6:

Figure 5. I would recommend removing the sentence “The effect is slightly stronger in low influence conditions.” At the end of the caption. Numerically that might be true, but this tiny difference is not meaningful. It seems not to be contributing to an argument anyway, doesn’t it? Also, I would advise to use it only if you tested the associations against each other also in the following figures and analyses.

Comment #7:

Figure 7. I was wondering why you use two different correction procedures for multiple comparisons and highlight these differently in the figure. I would suggest using either Bonferroni or FDR. The different number of stars for the different measures do not seem to make sense. Also, the legend is not easy to understand. The terms Necessary + predictive (why in italics) are only explained in the caption. I think you can make it easier for the reader to understand what is going on her and exactly clarify what necessary and predictive means.

Comment #8:

There seems to be missing a “what” in l 6 “…or affecting prior notions about exactly is achievable or what we can or cannot do.”

**Have all data underlying the figures and results presented in the manuscript been provided?**

Reviewer #1: None

Reviewer #2: Yes

Reviewer #3: Yes

Reviewer #4: None

PLOS authors have the option to publish the peer review history of their article (what does this mean?). If published, this will include your full peer review and any attached files.

Reviewer #1: **Yes: **Anand K. Rampadarath

Reviewer #2: No

Reviewer #3: No

Reviewer #4: No
---

## [Decision Letter · Decision Letter 1]

31 Jan 2021

Dear Prof. Dayan,

Thank you very much for submitting your manuscript "Internality and the Internalisation of Failure: Evidence From a Novel Task" for consideration at PLOS Computational Biology.

As with all papers reviewed by the journal, your manuscript was reviewed by members of the editorial board and by several independent reviewers. In light of the reviews (below this email), we would like to invite the resubmission of a significantly-revised version that takes into account the reviewers' comments.

Thank you for revising the manuscript. We have collected comments from three reviewers. For the interest of faster dessimination of research findings, we are moving forward with the existing reviews for you to revise. Please note that Reviewer 4's comment might come back later, in which case we will send to you to address.

Review # 2 has requested further analysis and clarification based on your revision, and made suggestion to the discussion. I hope you can address these concerns.

We cannot make any decision about publication until we have seen the revised manuscript and your response to the reviewers' comments. Your revised manuscript is also likely to be sent to reviewers for further evaluation.

Sincerely,

Ming Bo Cai

Associate Editor

PLOS Computational Biology

Daniele Marinazzo

Deputy Editor

PLOS Computational Biology

Thank you for revising the manuscript. We have collected comments from three reviewers. For the interest of faster dessimination of research findings, we are moving forward with the existing reviews for you to revise. Please note that Reviewer 4's comment might come back later, in which case we will send to you to address.

Review # 2 has requested further analysis and clarification based on your revision, and made suggestion to the discussion. I hope you can address these concerns.

Reviewer's Responses to Questions

**Comments to the Authors:**

Reviewer #1: Reproducibility report has been uploaded as an attachment.

Reviewer #2: The authors have made a substantial effort to incorporate the reviewers’ feedback into their manuscript, which we believe has improved the paper by further validating the relationship between their task and model. While we believe that this is a strong paper about a novel approach to studying complex decision making, their validations have raised some further questions about their best-performing model, that should be addressed in order to have full confidence in their conclusions.

1. We commend the authors for testing whether the participants are learning the value of individual features in the task. However, the feature-learning we had in mind involved learning the value of multiple features simultaneously, instead of the heuristic approach of just tracking a single feature. Given that two of the 1-feature RW models would be the second and third best performing models, it’s worth investigating whether participants learn the value of multiple features.

a) The authors should test whether 2- or 3-feature RW models out-perform the vehicle-level RW model.

b) Given the strong performance of even 1-feature RW models, the authors should consider reporting this approach in the manuscript.

2. We need some clarification on the author’s implementation of the ideal observer analysis. Our understanding of the optimal behavior is that it should be closest to the additive model (e.g., the RW models are framed as model-free), for which you have low confusability with the RW models. However, it appears that the RW fit to the ideal observer produces substantial vehicle-learning components.

If our understanding is correct, this may have several important implications for how we understand these learning models. The authors should try to understand why the ideal observer appears to use a model-free learning strategy, including the recommendations below. It may also be the case that the optimal strategy involves the use of vehicle-level learning. If this is the ture, the authors would do well to understand why this is the case, and ought to reframe these learning processes as part of the optimal decision strategy, instead of a model-free heuristic as they currently present in the text.

a) It could be that there are parameter trade-offs that allow RW components to mimic other parameterizations. The authors should perform parameter recovery on their winning model to ensure that they can accurately identify the contributions of the RW learning parameters.

b) There may be something about the trial sequence that confounds vehicle-level learning with the optimal decision strategy. If this is plausible, the authors should validate this by fitting the ideal observer under different trial sequences.

3. We still believe that the authors over-interpreting the implications of these results for the functional role of VMPFC. We suggest the reviewers limit their discussion of VMPFC to a discussion of future directions, for example suggesting that VMPFC may play a role in decision making in this task. The connection between controllability, aversive learning, and VMPFC is too speculative, given the nature of the data in this experiment.

Even then, the RW model suggests that participants placed very little weight on controllability, and much more on reward, distance, and especially value-learning. To the extent that the implications for VMPFC depend on controllability, the authors should be cautious in generalizing the results from this experiment. Similarly, the authors should consider whether situating this paper within the controllability literature in their introduction does service to the cognitive strategies that appear to be employed in this task.

Reviewer #3: The authors have answers my concerns and I think the paper is now improved.

**Have all data underlying the figures and results presented in the manuscript been provided?**

Reviewer #1: None

Reviewer #2: Yes

Reviewer #3: Yes

PLOS authors have the option to publish the peer review history of their article (what does this mean?). If published, this will include your full peer review and any attached files.

Reviewer #1: No

Reviewer #2: No

Reviewer #3: No
---

## [Decision Letter · Decision Letter 2]

19 May 2021

Dear Prof. Dayan,

Thank you very much for submitting your manuscript "Internality and the Internalisation of Failure: Evidence From a Novel Task" for consideration at PLOS Computational Biology. As with all papers reviewed by the journal, your manuscript was reviewed by members of the editorial board and by several independent reviewers. The reviewers appreciated the attention to an important topic. Based on the reviews, we are likely to accept this manuscript for publication, providing that you modify the manuscript according to the review recommendations.

We are very glad to have received your revision. Reviewers are happy with the updated manuscript. There are a few suggestions from Reviewer #2, which we hope you can incorporate.

Sincerely,

Ming Bo Cai

Associate Editor

PLOS Computational Biology

Daniele Marinazzo

Deputy Editor

PLOS Computational Biology

[LINK]

We are very glad to have received your revision. Reviewers are happy with the updated manuscript. There are a few suggestions from Reviewer #2, which we hope you can incorporate.

Reviewer's Responses to Questions

**Comments to the Authors:**

Reviewer #1: Reproducibility report has been uploaded as an attachment.

Reviewer #2: The authors have diligently replied to our concerns, and we are satisfied (and impressed) with the contents of this paper, with some amendments to how the uncertainties raised by parameter recovery should be conveyed.

We recommend softening the tone in interpreting the recovery of the learning-from-losses (epsilon_l) parameter: it shows strong posterior correlations with other parameters, and the recovery is on the low side, especially given the small sample size. More conservatively conveying your certainty in these analyses, in conjunction with the parameter recoveries you report in the supplement, will informative for a reader whose primary focus are these individual differences. Similarly, the caption for figure 6 should mention these difficulties in recovery.

Please also match the order of the parameters in FigS5 to FigS2, relabel the ‘gamma’ parameter in FigS5 to ‘alpha_H’, and include the parameter names in the figure caption.

**Have the authors made all data and (if applicable) computational code underlying the findings in their manuscript fully available?**

Reviewer #1: None

Reviewer #2: **No: **Data are available, code is not

PLOS authors have the option to publish the peer review history of their article (what does this mean?). If published, this will include your full peer review and any attached files.

Reviewer #1: No

Reviewer #2: No

Figure Files:

Data Requirements:

Reproducibility:

References:

---

## [Editor Report · Decision Letter 3]

31 May 2021

Dear Prof. Dayan,

We are pleased to inform you that your manuscript 'Internality and the Internalisation of Failure: Evidence From a Novel Task' has been provisionally accepted for publication in PLOS Computational Biology.

Best regards,

Ming Bo Cai

Associate Editor

PLOS Computational Biology

Daniele Marinazzo

Deputy Editor

PLOS Computational Biology

Editor additional comment:

Since the data and code are small in size, you can either keep them as supplementary material or upload to a repository. If you choose the latter, please ensure the link is included in the manuscript when finalizing it for print. As a very minor but easily fixable issue mentioned by reproducibility review last time, please fix the path to be relative path in your code so that readers can run your code without having to modify path by themselves.

---

## [Editor Report · Acceptance letter]

2 Jul 2021

PCOMPBIOL-D-20-01109R3 

Internality and the Internalisation of Failure: Evidence From a Novel Task

Dear Dr Dayan,

I am pleased to inform you that your manuscript has been formally accepted for publication in PLOS Computational Biology. Your manuscript is now with our production department and you will be notified of the publication date in due course.

With kind regards,

Agota Szep
